# ACTION-GUIDED ATTENTION FOR VIDEO ACTION ANTICIPATION

**Tsung-Ming Tai**[1,2]**, Sofia Casarin**[1]**, Andrea Pilzer**[2]**, Werner Nutt**[1]**, Oswald Lanz**[1]
[1] Free University of Bozen-Bolzano
[2] NVIDIA
`{ntai,apilzer}@nvidia.com`
`{casarin,nutt,lanz}@inf.unibz.it`

## ABSTRACT

Anticipating future actions in videos is challenging, as the observed frames provide only evidence of past activities, requiring the inference of latent intentions to predict upcoming actions. Existing transformer-based approaches, which rely on dot-product attention over pixel representations, often lack the high-level semantics necessary to model video sequences for effective action anticipation. As a result, these methods tend to overfit to explicit visual cues present in the past frames, limiting their ability to capture underlying intentions and degrading generalization to unseen samples. To address this, we propose Action-Guided Attention (AGA), an attention mechanism that explicitly leverages predicted action sequences as queries and keys to guide sequence modeling. Our approach fosters the attention module to emphasize relevant moments from the past based on the upcoming activity and combine this information with the current frame embedding via a dedicated gating function. The design of AGA enables post-training analysis of the knowledge discovered from the training set. Experiments on the widely adopted EPIC-Kitchens-100 benchmark demonstrate that AGA generalizes well from validation to unseen test sets. Post-training analysis can further examine the action dependencies captured by the model and the counterfactual evidence it has internalized, offering transparent and interpretable insights into its anticipative predictions.

## 1 INTRODUCTION

Anticipating future actions in videos is challenging in computer vision, with broad implications for assistive systems, robotics, autonomous vehicles, and interactive entertainment. The core difficulty stems from the need to predict upcoming actions based solely on observed video frames, which provide only indirect evidence of subtle human intentions. Unlike action recognition, where its annotations are present and synchronized with observed frames, action anticipation, on the other hand, requires inferring future action labels from up-to-current observations. The observed frames often contain merely partially revealed and ambiguous visual cues, and the fact that the same past observations could lead to multiple possible future outcomes further makes the task inherently uncertain.

The emergence of the vision transformer has appeared as a dominant paradigm in video action anticipation modeling. However, existing vision transformer approaches typically rely on conventional self-attention over the token representation transformed from the video pixels. This design principle could have advantages in the action recognition task, as self-attention can effectively construct the visual patterns and match their synchronized annotations. Nonetheless, the inherent non-deterministic nature of the future makes action anticipation a more complex task than pattern recognition. Conventional self-attention can therefore be misled by visual clutter, potentially leading to overfitting.

To address this problem, we propose leveraging action predictions to sequentially guide the focus of attention. We call our method Action-Guided Attention (AGA), where both queries and keys are represented by action probabilities. The design builds upon the fundamentals of dot-product attention, which attends to values through correlations between query and key representations. Instead of relying on pixel-level features, AGA uses predicted actions as high-level semantic guidance. This

formulation explicitly models the idea of predicting the next action conditioned on previously predicted actions, learning dependencies among actions to improve the anticipation of future events. The output of the AGA aggregates relevant past moments and further adaptively mixes the information derived from the current frame inputs through a dedicated gating function to balance between history context and current evidence.

On EPIC-Kitchens-100, AGA generalizes effectively from validation to unseen test sets, with a consistently narrow gap that suggests resistance to overfitting on partially observed video inputs. Its robustness is further validated on EPIC-Kitchens-55 and the sparsely annotated EGTEA Gaze+.

Moreover, the design of AGA also enables post-training analysis to uncover the knowledge learned during training. Through forward analysis and backward analysis, we can examine the action dependencies captured by the model and the counterfactual evidence it has internalized, offering transparent and interpretable insights into its anticipative predictions.

## 2 RELATED WORK

A broad spectrum of anticipation methods have been proposed over the past five years. Sequence modeling approaches, primarily based on recurrent networks (e.g., RNN, GRU, LSTM), were initially explored to map observed features to future actions. RULSTM (Furnari & Farinella, 2020) employs two LSTMs, the first summarizes the observed video sequence, while the second unrolls predictions according to the temporal distance to the start of the future action. By explicitly incorporating the anticipation interval and unrolling toward the end of the interval, this architecture established a strong baseline on large-scale video action anticipation benchmarks such as EPIC-Kitchens-55 (Damen et al., 2018) and EPIC-Kitchens-100 (Damen et al., 2022). However, sequential recurrent models are prone to error accumulation as time progresses. To address this, SRL (Qi et al., 2021) introduces a regulation mechanism that emphasizes novel information at each timestamp, contrasting it with previously observed content and modeling its correlation with past frames, thereby improving upon the RULSTM baseline. Building on this line, ImagineRNN (Wu et al., 2020) further enhances unrolling by learning to explicitly predict frame-wise differences within the anticipation interval in a contrastive manner.

More recently, transformer-based architectures have emerged, delivering significant improvements in action recognition and anticipation. The Anticipative Visual Transformer (AVT) (Girdhar & Grauman, 2021) introduced a causal attention decoder on top of the standard Vision Transformer framework for action anticipation, achieving state-of-the-art performance at its time of publication. MemViT (Wu et al., 2022) extended AVT by storing longer historical context within the attention keys and values through token compression techniques. Later, RaftFormer (Girase et al., 2023) addressed the high computational cost of attention by optimizing the model design for efficiency and faster inference.

Besides, several works have explored multi-modal integration to enhance prediction accuracy. S-GEAR (Diko et al., 2024) emphasized semantic representations by jointly supervising visual and language branches. AFFT (Zhong et al., 2023) adapted the GPT-2 architecture and proposed an efficient fusion mechanism capable of incorporating additional modalities such as audio and optical flow. InAViT (Roy et al., 2024) leveraged prior information in the form of hand masks to disentangle human interactions from environmental clutter. More recent directions investigate the use of vision-language models and large language models for anticipation (Zhang et al., 2023; Mittal et al., 2024; Wang et al., 2025). Other extensions include semantic action augmentation (Qiu & Rajan, 2025) and long-term anticipation (Zhao et al., 2023; Zatsarynna et al., 2025), further broadening the scope of multi-modal anticipation research.

Prior work also explores the semantic label space in anticipation, but unlike AGA they do not feed predictions back into the attention mechanism to reweight the visual representation. Abu Farha et al. (2018) use RNN/CNN predictors to decode future actions in the label space. Ke et al. (2019) refine predictions by injecting temporal features into an attention module. Zhao & Wilde (2020) condition the initial visual observation on an action-time sequence through a conditional GAN to generate diverse futures. However, in all these methods the predictions remain separate from feature extraction, they do not guide the attention over visual tokens. AGA is distinct in that it uses the

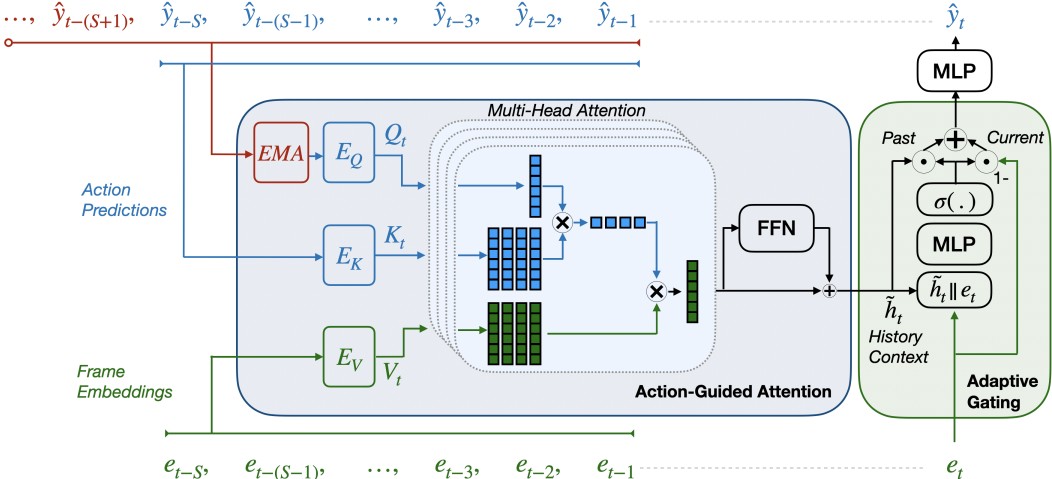

Figure 1: **Architecture Overview.** The model consists of two modules. The *Action-Guided Attention* uses the most recent $S$ action predictions as keys, the exponential moving average (EMA) of all predicted actions as the query, and $S$ frame embeddings as values to generate a history context $\tilde{h}_t$. The *Adaptive Gating* then integrates this history context with the current frame embedding $e_t$ to produce a fused representation, which is mapped to the new prediction $\hat{y}_t$.

model's own evolving action predictions to dynamically focus attention on semantically meaningful visual cues.

In this work, we focus on a fundamental challenge in video action anticipation, which causes overfitting due to incomplete and uncertain observations associated with the future action. We propose an alternative attention design, AGA, that incorporates high-level, task-guided action representations as queries and keys, thereby mitigating attention to over-reliance on unreliable visual cues from partial video evidence. Our study considers explicitly anticipation methods that use RGB video frames as input, excluding those that integrate multi-modal or auxiliary knowledge sources such as text.

## 3 METHOD

Let $(x_t)_{t \geq 0}$ be a sequence of video frames sampled every $\Delta t > 0$ seconds, where $x_t \in \mathbb{R}^{C \times H \times W}$. For indices $r \leq s$, we write $x_{r:s}$ to denote the subsequence $(x_{\max(0,r)}, \ldots, x_s)$, that is, values with $r < 0$ are clipped at zero; if $s$ exceeds the total sequence length then set it to the last index, and $s \geq 0$. Here, $t$ denotes the discrete frame index, corresponding to real time $t \cdot \Delta t$. Given the observed frames $x_{0:T}$, the goal is to predict the future action $y_{T+t_a} \in \mathbb{R}^{N_c}$, represented as a one-hot vector over $N_c$ classes, which occurs $t_a \cdot \Delta t$ seconds after the last observed frame. For each index $0 < t < T$, the model outputs a probability distribution $\hat{y}_t \in [0, 1]^{N_c}$ as an estimate of the true future action $y_{t+t_a}$.

### 3.1 ACTION-GUIDED ATTENTION

We introduce a conditional attention mechanism that extends the standard formulation of Vaswani et al. (2017) by conditioning queries ($Q$) and keys ($K$) on semantic-level action predictions, while using frame-encoded video features as values ($V$). Ideally, the ground-truth action label $y_t$ would provide the most reliable signal for guiding the estimation of future actions. However, because the ground-truth is unavailable at inference time, we instead rely on the self-predicted distribution $\hat{y}_t$ as an approximate guide. In dot-product attention, correlations between query and key yield weights that select relevant values. Our design represents the features of a video frame by its corresponding action prediction, and then utilizes the dot-product between prediction embeddings to aggregate relevant frame features to forecast the anticipated action. The overall architecture is shown in Figure 1.

At each timestep $t$, the input video frame $x_t$ is first processed by a frame-based backbone $f_{\text{backbone}}(\cdot)$, followed by an trainable encoder $f_x(\cdot)$ that extracts the frame features $e_t$:

$$e_t = f_x(f_{\text{backbone}}(x_t)).$$

The trainable function $f_x(.)$ finetunes features from a frozen backbone, keeping the backbone modular and interchangeable for any image architecture.

These features are subsequently stored in a first-in, first-out (FIFO) queue of size $S$, a hyperparameter that determines the temporal window over which the model can reference past information. The number of the keys and values is thus dependent on $S$.

At each timestep, the frame features stored in the queue serve as the values, while the predictions from the same time, which can be viewed as semantic tags summarizing the information up to the current moment, are used to construct the keys. Specifically, instead of using the perceptual features $e$, we employ the past predictions $\hat{y}$ to represent high-level semantics in the attention mechanism. Formally, we define:

$$K_t = E_K(\hat{y}_{t-S:t-1}) \qquad V_t = E_V(e_{t-S:t-1}), \tag{1}$$

where $E_K(\cdot)$ and $E_V(\cdot)$ are implemented as multilayer perceptrons (MLP).

Correspondingly, the query also leverages the sequence of action predictions to match the semantic level with the keys. To effectively incorporate longer temporal dependencies, we apply an exponential moving average (EMA) over the past action predictions to form the query:

$$\bar{y}_t = \alpha\hat{y}_{t-1} + (1-\alpha)\bar{y}_{t-1} \qquad Q_t = E_Q(\bar{y}_t), \tag{2}$$

where $\bar{y}_0$ is initialized as the zero vector and $\alpha$ is empirically set to 0.8 (see Section 4.3 and Appendix B.1.2 for experiments on the choice of $\alpha$). The function $E_Q(\cdot)$ consists of an MLP. Note that $K_0$ and $V_0$ are undefined and are never used for attention. When the queue is empty, the first input frame, the attention is bypassed and its output is initialized to a zero vector. Starting from the second frame, the queue contains at least one timestep for both $Q$ and $K$, allowing the attention mechanism to begin processing information even if the queue is not yet full.

Then, with $Q_t$, $K_t$, and $V_t$ defined, the multi-head dot-product attention is applied:

$$h_t = \text{MultiHead}(Q_t, K_t, V_t) = (\text{head}_1 \| \cdots \| \text{head}_h)W^o,$$

where

$$\text{head}_i = \text{Softmax}\left( \frac{(Q_t W_i^Q)(K_t W_i^K)^\top}{\sqrt{d}} \right)(V_t W_i^V). \tag{3}$$

Here, $h_0$ is initially set to a zero-vector, $W_i^Q, W_i^K, W_i^V$ are the trainable parameters for multi-heads, and $W^o$ is the output weight for aggregating the multi-head attention outputs.

As usual, the output of the attention module is further transformed by a feedforward network (FFN), implemented as an MLP, with a residual connection. The FFN complements the attention mechanism by introducing non-linear transformations, while the learnable parameters within the attention are confined to the input and output embedding projections:

$$\tilde{h}_t = h_t + \text{FFN}(h_t).$$

For both multi-head attention and FFN the normalization is performed in PreNorm style (Xiong et al., 2020) using the RMSNorm layer (Zhang & Sennrich, 2019). The outputs of the multi-head attention and the FFN are then fused with $e_t$ through the Adaptive Gating.

## 3.2 ADAPTIVE GATING

The relevance of the history context and the current visual evidence vary over time. Therefore, we componentwise fuse the history $\tilde{h}_t$ and evidence $e_t$ using a gating vector $g_t \in [0,1]^d$ that has the same dimension as $\tilde{h}_t$ and $e_t$, resulting in the output

$$o_t = g_t \odot \tilde{h}_t + (1 - g_t) \odot e_t,$$

where $\odot$ denotes componentwise multiplication. The gating vector itself is computed as $g_t = \sigma(\text{MLP}(\tilde{h}_t \parallel e_t))$, where the MLP is composed of two linear layers and a ReLU activation function while $\sigma(\cdot)$ denotes the sigmoid function. Adaptive Gating is illustrated in Figure 1.

Entries of $g_t$ close to 1 emphasize the history in those components, while entries near 0 favor the current visual evidence. This adaptive mixing lets the model rely on past or presence as needed.

Finally, the new prediction $\hat{y}_t$ is obtained from the Adaptive Gating output $o_t$ as

$$\hat{y}_t = \text{Softmax}(\text{MLP}(o_t)),$$

by applying an MLP, consisting of two linear layers with a ReLU activation to produce the logits, followed by a Softmax.

### 3.3 FORWARD ANALYSIS

The design of AGA gives us insights into how a trained model works. Specifically, we can answer questions of the kind: Given a currently observed action, which past actions does the model consider as relevant for predicting the next action? We refer to this as forward analysis.

For example consider the actions $oc = open\ cupboard$ and $cc = close\ cupboard$. Suppose that in the current frame the cupboard is opened and the history contains both, frames where the cupboard is opened and where it is being closed. Which of those are more relevant? To find out we create a query for the current observation, a key for each candidate action, and then compute the weights that the attention mechanism gives to each candidate with regard to the query.

More precisely, for each action $a$, let $y_a$ be the one-hot distribution with $y_a(a) = 1$ and $y_a(c) = 0$ for all action classes $c \neq a$. In our example, we create for $oc$ the query embedding $Q_{oc} = E_Q(y_{oc})$ and for $oc$ and $cc$ together the key embedding $K_{oc,cc} = E_K(y_{oc} \parallel y_{cc})$. Using the Softmax expression in Equation 3, we compute for each head $i$ a vector $w^{(i)} = (w_{oc}^{(i)}, w_{cc}^{(i)})$ of attention weights and then take the average $\bar{w} = 1/h \sum_{i=1}^{h} w^{(i)}$ of these vectors. The component $\bar{w}_{oc}$ reflects the average relevance given to frames where the cupboard is opened and $\bar{w}_{cc}$ where it is closed. Figure 2 shows actual weights for this example obtained from an AGA model trained on EPIC-Kitchens-100 dataset.

### 3.4 BACKWARD ANALYSIS

While forward analysis explains which past predictions the model *actually* relies on when producing its output, backward analysis instead asks a counterfactual question: If the next action were $a$, what changes in the past predictions would make the model assign a high probability to $a$?

Let $Y = (\hat{y}_{t-S}, \ldots, \hat{y}_{t-1})$ be the sequence of past predictions and let $\hat{y}_t = f_\theta(Y, X)$ denote the predicted distribution at time $t$ given the sequence of frames $X = X_{t-S:t}$. During training, each $\hat{y}_i$ is a probability distribution, but as a function $f_\theta$ is defined for arbitrary real vectors $Y$. Hence, for fixed $X$, the following expressions are well defined for any $Y$.

Suppose the model has produced the prediction $\hat{y}_t$, while the correct next action is $a$, represented by the one-hot distribution $y_a$. The discrepancy between prediction and target is measured by the cross-entropy

$$H(y_a, \hat{y}_t) = -\sum_c y_a(c) \log \hat{y}_t(c) = -\log \hat{y}_t(a),$$

where the sum is taken over all class labels $c$. Since $\hat{y}_t$ depends on the past predictions $Y$, we can ask how $Y$ can be changed so that the resulting prediction comes close to $y_a$. To capture this, we define

$$L(Y) = H\big(y_a, f_\theta(Y, X)\big) = -\log\big(f_\theta(Y, X)(a)\big).$$

Minimizing $L(Y)$ amounts to finding past predictions that make the target action $a$ more probable for the given model. We can find an approximate local minimum by gradient descent, choosing a step size $\eta$ and starting from the original predictions $Y^{(0)}$, iterating through

$$Y^{(j+1)} = Y^{(j)} - \eta \nabla_Y L(Y^{(j)}),$$

until the loss function plateaus, that is,

$$|L(Y^{j+1}) - L(Y^j)| < \epsilon$$

| Methods | Modality | Overall Classes | | | Unseen Classes | | | Tail Classes | | |
|---|---|---|---|---|---|---|---|---|---|---|
| | | Action | Verb | Noun | Action | Verb | Noun | Action | Verb | Noun |
| RULSTM (Furnari & Farinella, 2020) | RGB, Flow, Obj | 11.2 | 25.3 | 26.7 | 9.7 | 19.4 | 26.9 | 7.9 | 17.6 | 16.0 |
| AVT+ (Girdhar & Grauman, 2021) | RGB, Obj | 12.6 | 25.6 | 28.8 | 8.8 | 20.9 | 22.3 | 10.1 | 19.0 | 22.0 |
| AVT++ (Girdhar & Grauman, 2021) | RGB, Flow, Obj | 16.7 | 26.7 | 32.3 | 12.9 | 21.0 | 27.6 | 13.8 | 19.3 | 24.0 |
| AFFT-TSN+ (Zhong et al., 2023) | RGB, Flow, Obj, Audio | 13.4 | 19.4 | 28.3 | 9.9 | 14.0 | 24.2 | 10.9 | 12.0 | 19.5 |
| AFFT-Swin+ (Zhong et al., 2023) | RGB, Flow, Obj, Audio | 14.9 | 20.7 | 31.8 | 12.1 | 16.2 | 27.7 | 11.8 | 13.4 | 23.8 |
| RAFTformer (Girase et al., 2023) | RGB | 14.7 | 27.4 | 34.0 | - | - | - | - | - | - |
| RAFTformer-2B (Girase et al., 2023) | RGB | 15.4 | 30.1 | 34.1 | - | - | - | - | - | - |
| S-GEAR (Diko et al., 2024) | RGB, Obj | 14.7 | 25.9 | 32.0 | - | - | - | - | - | - |
| S-GEAR-2B (Diko et al., 2024) | RGB, Obj | 15.3 | 25.5 | 31.7 | - | - | - | - | - | - |
| S-GEAR-4B (Diko et al., 2024) | RGB, Obj | 15.5 | 26.6 | 32.6 | - | - | - | - | - | - |
| **AGA (Ours, Swin-B)** | RGB | **16.9** | **30.8** | **36.4** | **13.5** | **22.3** | **30.0** | **14.9** | **25.8** | **30.0** |

Table 1: **EK100 Test Results**. Evaluation on a unseen test set via the official challenge server.

for a threshold $\epsilon$, or when the maximum number of iteration steps *iter* has been exceeded, that is, $j \geq iter$.

The resulting $Y^*$ need not be itself a valid sequence of probability distributions, but it can still be decomposed into vectors as long as distributions and we can compare it with the actual distributions in $Y^{(0)}$. In the difference $\Delta Y = Y^* - Y^{(0)}$ positive entries indicate supportive changes and negative entries indicate blocking changes for the target action $a$. This provides a counterfactual perspective on the dependencies captured by the model, complementary to forward analysis.

Qualitative results of both forward and backward analyses are presented in the experiments section, where they provide concrete insights into the model decision-making process and the knowledge that our model can explore in the training dataset.

# 4 EXPERIMENTAL RESULTS

## 4.1 EXPERIMENTAL SETUP

We evaluated our proposed method using the TSN (Wang et al., 2016) and Swin-B (Liu et al., 2021) backbones, with AGA applied on frozen features as a single attention layer with hidden size 2048 and 16 heads. The queue size $S$ is set to 16 (see Appendix B.1.1 for experiments on the choice of $S$).Models were trained for 50 epochs with a batch size of 128, using AdamW with a learning rate of $2 \times 10^{-4}$, weight decay of 0.01, and a cosine schedule. All input frames were resized to 224×224. Performance was measured using Mean Top-5 Recall (MT5R), which averages recall across classes to mitigate the dominance of frequent actions. Following the training protocol of (Girdhar & Grauman, 2021), we compute class-reweighting factors with the inverted class frequency, and apply these weights when supervising the anticipation predictions with the ground-truth annotations.Results were reported at the dataset-specific anticipation interval $\tau_a$. We converted $\tau_a$ (provided in seconds) into the corresponding anticipation index $t_a$ (used by our method) as the number of time units $\Delta t$ to reach $\tau_a$, that is, $\tau_a = t_a \cdot \Delta t$.

Experiments were conducted on three benchmarks. **EPIC-Kitchens-100 (EK100)** (Damen et al., 2022) contains 100 hours of video with 3,806 actions, including 67,217 training and 9,668 validation segments, and was evaluated at $\tau_a = 1$s and $\Delta t = 1$s.**EPIC-Kitchens-55 (EK55)** (Damen et al., 2018) contains 55 hours of video with 2,513 actions, including 23,492 training and 4,979 validation segments, and was evaluated using Top-1 and Top-5 accuracy as well as MT5R at $\tau_a = 1$s and $\Delta t = 1$s.**EGTEA Gaze+** (Li et al., 2018) comprises 28 hours of egocentric video with 106 actions, including 8,299 training and 2,022 validation clips, where we followed the protocol in (Girdhar & Grauman, 2021) and reported Top-1 accuracy and mean Top-1 recall at $\tau_a = 0.5$s and $\Delta t = 0.5$s.

All the benchmark scores are evaluated on the time point $T \cdot \Delta t + \tau_a$, where $T$ is the sequence length.

## 4.2 RESULTS

Tables 1 and 2 present the performance of the proposed AGA in comparison with prior methods on EPIC-Kitchens-100 for the unseen test set and the validation set, respectively. On the validation

| Methods | Backbone | Overall Classes | | | Unseen Classes | | | Tail Classes | | |
|---|---|---|---|---|---|---|---|---|---|---|
| | | Action | Verb | Noun | Action | Verb | Noun | Action | Verb | Noun |
| RULSTM (Furnari & Farinella, 2020) | TSN | 13.3 | 27.5 | 29.0 | - | - | - | - | - | - |
| AVT-h (Girdhar & Grauman, 2021) | TSN | 13.6 | 27.2 | 30.7 | - | - | - | - | - | - |
| AVT-h (Girdhar & Grauman, 2021) | irCSN152 | 12.8 | 25.5 | 28.1 | - | - | - | - | - | - |
| AVT-h (Girdhar & Grauman, 2021) | AVT-b | 14.9 | 30.2 | 31.7 | - | - | - | - | - | - |
| AFFT (Zhong et al., 2023) | TSN | 16.4 | 21.3 | 32.7 | 13.6 | 24.1 | 25.5 | 14.3 | 13.2 | 25.8 |
| AFFT (Zhong et al., 2023) | Swin-B | 17.6 | 23.4 | 33.7 | 15.2 | 24.5 | 25.4 | 15.3 | 15.6 | 26.5 |
| MeMViT 16x4 (Wu et al., 2022) | MViTv2-16 | 15.1 | 32.8 | 33.2 | 9.8 | 27.5 | 21.7 | 13.2 | 26.3 | 27.4 |
| MeMViT 32x3 (Wu et al., 2022) | MViTv2-24 | 17.7 | 32.2 | 37.0 | 15.2 | 28.6 | 27.4 | 15.5 | 25.3 | 31.0 |
| RAFTformer (Girase et al., 2023) | MViTv2-16 | 17.6 | **33.3** | 35.5 | - | - | - | - | - | - |
| RAFTformer-2B (Girase et al., 2023) | MViTv2-16 + 24 | 19.1 | 33.8 | 37.9 | - | - | - | - | - | - |
| S-GEAR (Diko et al., 2024) | TSN | 14.9 | 25.8 | 29.8 | - | - | - | - | - | - |
| S-GEAR (Diko et al., 2024) | ViT-B | 18.3 | 31.1 | 37.3 | - | - | - | - | - | - |
| S-GEAR-2B (Diko et al., 2024) | ViT-B x2 | 19.6 | 32.7 | 37.9 | - | - | - | - | - | - |
| **AGA (Ours)** | TSN | 17.5 | 32.2 | 35.7 | 11.9 | 31.4 | 24.9 | 16.8 | 26.9 | 31.8 |
| **AGA (Ours)** | Swin-B | **18.8** | 32.5 | **38.7** | **16.3** | **34.4** | **28.5** | **18.4** | **27.4** | **35.0** |

Table 2: **EK100 Validation Results**. Comparison of methods on the validation set using only RGB input. Methods highlighted in gray use the same TSN backbone weights.

| Methods | Backbone | Top-1 Acc | Top-5 Acc | MT5R |
|---|---|---|---|---|
| RULSTM (Furnari & Farinella, 2020) | TSN | 13.1 | 30.8 | 12.5 |
| TempAgg (Sener et al., 2020) | TSN | 12.3 | 28.5 | 13.1 |
| ImagineRNN (Wu et al., 2020) | TSN | 13.7 | 31.6 | - |
| SRL (Qi et al., 2021) | TSN | - | 31.7 | 13.2 |
| AVT-h (Girdhar & Grauman, 2021) | TSN | 13.1 | 28.1 | 13.5 |
| AVT-h (Girdhar & Grauman, 2021) | AVT-b | 12.5 | 30.1 | 13.6 |
| AVT-h (Girdhar & Grauman, 2021) | irCSN152 | 14.4 | 31.7 | 13.2 |
| DCR (Xu et al., 2022) | TSN | 13.6 | 30.8 | - |
| DCR (Xu et al., 2022) | irCSN152 | 15.1 | 34.0 | - |
| DCR (Xu et al., 2022) | TSM | 16.1 | 33.1 | - |
| RAFTformer (Girase et al., 2023) | TSN | 13.8 | - | - |
| S-GEAR (Diko et al., 2024) | TSN | 15.6 | 32.8 | - |
| S-GEAR (Diko et al., 2024) | irCSN152 | 16.2 | 33.1 | - |
| S-GEAR (Diko et al., 2024) | ViT-B | 15.8 | 34.5 | - |
| **AGA (Ours)** | TSN | 13.5 | 32.1 | 14.3 |
| **AGA (Ours)** | Swin-B | **16.3** | **37.4** | **16.6** |

Table 3: **EK55 Validation Results**. Top-1/Top-5 action accuracy and MT5R at $\tau_a = 1s$. Methods highlighted in gray use the same TSN backbone weights.

set, all methods are evaluated using the RGB modality only, and approaches that exploit additional modalities or external knowledge are excluded to ensure fair comparison. With the TSN backbone (highlighted in gray background colors), AGA achieves 17.5% accuracy, outperforming AFFT at 16.4%, S-GEAR at 14.9%, and other baselines. Replacing TSN with the stronger Swin-B backbone further improves performance from 17.5% to 18.8%, with noticeable gains observed on unseen and tail classes. Methods that ensemble features from multiple backbones are highlighted in gray.

On the unseen test set, evaluated through the official challenge server, AGA maintains consistent performance, demonstrating strong generalization ability. Many methods listed in Table 1 are primarily competition submissions that rely on multi-modal inputs or ensemble strategies, whereas AGA achieves competitive scores without such enhancements. The gap between validation and test is consistently small, indicating that AGA generalizes effectively beyond the validation set compared with other methods. Leveraging action predictions to guide attention prevents the model from over-emphasizing the visual clutters, which is especially important in action anticipation where uncertainty is high and observations are less reliable than in other video recognition tasks.

Tables 3 and 4 compare the performance of AGA on two smaller-scale datasets, EK55 and EGTEA Gaze+. EGTEA Gaze+ provides sparse annotations only for the target action, without detailed labels for the preceding frames in the pre-action observation window. Accordingly, our model must rely entirely on its own predictions of prior activities without supervision. Nevertheless, empirical results show that AGA maintains strong performance, even under constrained annotation availability and sparse supervision. On EK55, S-GEAR reports higher accuracy than AGA under the same TSN backbone, likely due to its use of higher-resolution inputs (384×384 compared to 224×224).

| Methods | Top-1 Acc | | | Mean Top-1 Recall | | |
|---|---|---|---|---|---|---|
| | Action | Verb | Noun | Action | Verb | Noun |
| I3D-Res50 (Carreira & Zisserman, 2017) | 34.8 | 48.0 | 42.1 | 23.2 | 31.3 | 30.0 |
| FHOI (Liu et al., 2020) | 36.6 | 49.0 | 45.5 | 32.5 | 32.7 | 25.3 |
| TSN-AVT-h (Girdhar & Grauman, 2021) | 39.8 | 51.7 | 50.3 | 28.3 | 41.2 | 41.4 |
| AVT (Girdhar & Grauman, 2021) | 43.0 | 54.9 | 52.2 | 35.2 | **49.9** | 48.3 |
| TSN-AFFT (Zhong et al., 2023) | 42.5 | 53.4 | 50.4 | 35.2 | 42.4 | 44.5 |
| **AGA (Ours, TSN)** | 43.5 | 54.3 | 52.2 | 35.5 | 43.8 | 46.6 |
| **AGA (Ours, Swin-B)** | **45.4** | **55.9** | **54.3** | **37.4** | 46.5 | **49.3** |

Table 4: **EGTEA Gaze+ Validation Results**.Top-1 accuracies and Mean Top-1 Recall at $\tau_a = 0.5s$. Methods highlighted in gray use the same TSN backbone weights.

| Configuration | MT5R |
|---|---|
| Reference (LSTM) | 14.5 |
| Baseline (Causal Attention) | 15.9 |
| **Action-Guided Attention** | 18.2 |
| **Action-Guided Attention + Adaptive Gating** | **18.8** |

| EMA $\alpha$ | MT5R |
|---|---|
| $\alpha = 0.0$ | 17.4 |
| $\alpha = 0.2$ | 18.7 |
| $\alpha = 0.4$ | 18.4 |
| $\alpha = 0.6$ | 18.5 |
| $\alpha = \mathbf{0.8}$ | **18.8** |
| $\alpha = 1.0$ | 18.2 |

Table 5: Ablation study of proposed Action-Guided Attention.    Table 6: Selection of EMA $\alpha$ for $Q$.

Moreover, EK55 evaluation is majorly reported in overall accuracy rather than in class-mean recall, which biases results toward common classes and obscures performance on rare actions.

### 4.3 ABLATION STUDY

Table 5 presents the ablation study of the proposed model. The analysis begins with a baseline model, causal attention (e.g., AVT). Replacing the standard dot-product attention with queries and keys defined by action predictions yields a substantial improvement, raising accuracy from 15.9% to 18.2%. In this setting, adaptive gating is disabled, so the attention output is directly added to the current frame input without weighting past and current features. Incorporating adaptive gating provides additional gains, further increasing accuracy from 18.2% to 18.8%. For reference, an LSTM baseline trained under the same conditions is also reported.

Table 6 examines the impact of different $\alpha$ values in the exponential moving average used to form the query. Larger values of $\alpha$ make the query depend more heavily on the most recent prediction. When $\alpha = 0$, the query collapses to a constant (projected from the zero vector) across all timesteps, leaving attention unconditioned and driven solely by the keys, which results in poor anticipation accuracy. For $0.2 \leq \alpha \leq 0.8$, the query aggregates action dynamics over time, providing a coherent action-guided signal and improving performance. When $\alpha = 1$, performance again drops because the query depends only on the latest prediction. Based on empirical results, we adopt $\alpha = 0.8$ in our model. All ablations reported here were conducted on the EK100 validation set and evaluated using MT5R. Appendix B.1.2 informs about studies on the other two datasets used in the experiments.

### 4.4 FORWARD ANALYSIS

Figure 2 illustrates the attention focus on past actions given different queries. Since AGA treats past actions (keys) as an unordered set, the order shown along the x-axis is not indicative. Four examples are analyzed using *open/close cupboard* and *open/close fridge* as queries to examine how the model allocates attention across past actions.

In the first two examples, where the queries are *open cupboard* and *close cupboard*, the model consistently attends more on earlier *open cupboard* rather than *close cupboard*. This behavior suggests that the model learns to revisit earlier events and search for evidence that indicates the duration of an *open cupboard* state, which may not be visible when the cupboard is closed. Moreover, the attention distribution conditioned on *open cupboard* is more uniform, implying that the next action following an *open* event is less certain and can be supported by multiple cues. By contrast, conditioning on

*close cupboard* produces a relatively sharper distribution, reflecting more substantial confidence that the next action is linked to objects accessed during the preceding open period.

In the third and fourth examples, where the queries are *open fridge* and *close fridge*, the model shows a similar pattern. Conditioning on *open fridge* yields a more uniform attention across past actions, though food-related interactions receive higher weights. When conditioned on *close fridge*, the attention becomes more concentrated, focusing primarily on *take food*, indicating that the model leverages this evidence to predict subsequent actions involving the *food* that has been taken before. Overall, these examples suggest that AGA captured plausible dependencies and learnt meaningful associations between actions to focus on scenes in the past relevant for the immediate future.

### 4.5 BACKWARD ANALYSIS

Figure 3 presents the backward analysis of the model trained on the EK100 dataset.

The example is taken from the EK100 validation set. The original top-5 predicted actions are shown along with the counterfactual top-5 supportive actions via proposed backward analysis optimized toward the target action *take pan* (which originally ranked 5th). Each column corresponds to a timestep, with the final column (gray background) representing the anticipated output of the video clip. Actions marked in red indicate those suppressed by backward analysis, while those in green indicate promoted actions.

In the original top-5 predictions, the model prioritized actions such as *eat squash*, *dry ladle*, and *close cupboard*, with *take pan* largely absent. This pattern reveals a tendency to favor immediate, contextually dominant actions, as reflected in the diversity of verbs and nouns. Backward analysis conditioned on the counterfactual target *take pan* discovers an alternative reasoning path. Supportive actions involving containers and utensils, such as *take spatula*, *put pan*, *insert pan*, and *take ladle*, forming a semantically coherent trajectory toward the target. These results, obtained with model weights frozen, suggest that the model has already encoded multiple plausible futures. However, when faced with broad uncertainty, it originally distributed focus across diverse environmental cues. Backward analysis highlights a counterfactual path that converges on utensils, offering insights into both the model inference process and the latent knowledge it has acquired.

Backward analysis in the paper was conducted using a stopping criterion of $\epsilon = 1\text{e-}6$, a step size of $\eta = 1\text{e}2$, and a maximum of $iter = 5000$. Under this configuration, the optimization consistently identified counterfactual actions within the top-10 predictions as the new top-1 choice, validated across 30 validation samples drawn from EK100. When further varying the step size from 1e-1 to 1e5, we found that the same behavior remained stable for step sizes in the range 1e1 to 1e4, using the same stopping criterion.

### 4.6 ADAPTIVE GATING

Figure 4 shows two action sequences together with their average gating ratios, retrieved from the adaptive gating module during inference. Each sequence contains colored segments representing

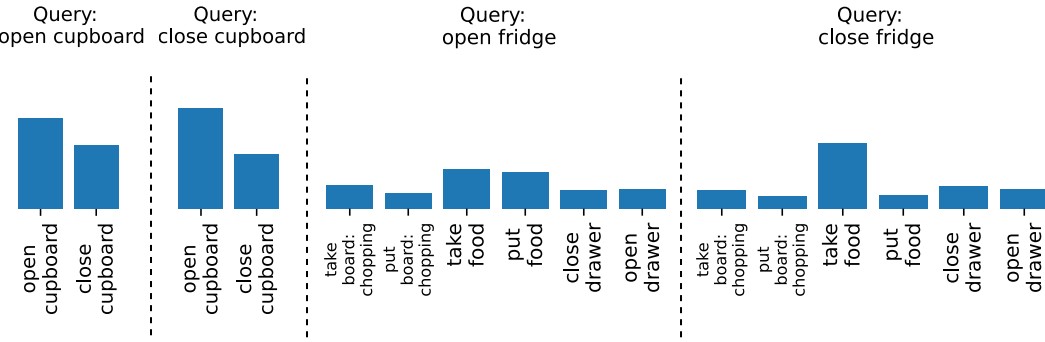

Figure 2: **Forward Analysis** identifies which past actions the model attends to when predicting its next action in response to a query. This analysis was conducted on the model trained with EK100.

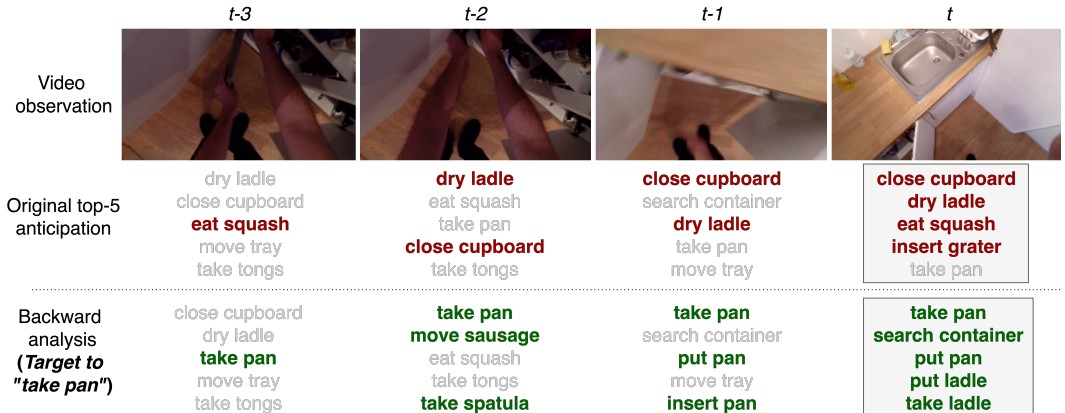

Figure 3: **Backward Analysis.** The figure compares the top-5 original predicted actions with the counterfactual supportive actions optimized toward the target action *take pan*. Each column represents a timestep; the final column shows the anticipated output. Suppressed actions are highlighted in red and promoted actions appear in green. This example is drawn from the EK100 validation set.

actions and black segments representing background or non-action content. The adaptive gate balances historical and current information: higher values emphasize past context, while lower values prioritize present input. Empirical observations reveal that fluctuations in the gating value often correlate with action transitions. For instance, the gate tends to rise during background segments to preserve past information, and then shifts attention to the current input when an action occurs. Importantly, the adaptive gate is not explicitly supervised by action boundaries; its context-aware behavior emerges naturally from training on the anticipation task.

## 5 CONCLUSION

We present AGA, Action-Guided Attention for video action anticipation, which leverages past predictions to create semantic representations that guide the model's attention to parts of the history relevant for anticipating the immediate future. In addition, AGA dynamically mixes past and present visual information with its adaptive gating mechanism. Experiments show resistance to overfitting on top of overall competitive performance. Furthermore, AGA enables post-training forward and backward analysis, offering insight into the learned action dependencies and the reasoning process behind anticipation.

## 6 REPRODUCIBILITY STATEMENT

Implementation details are presented in Sections 3, 4.1, and Appendix B. The source code and processing scripts are publicly available at: `https://github.com/CorcovadoMing/AGA`

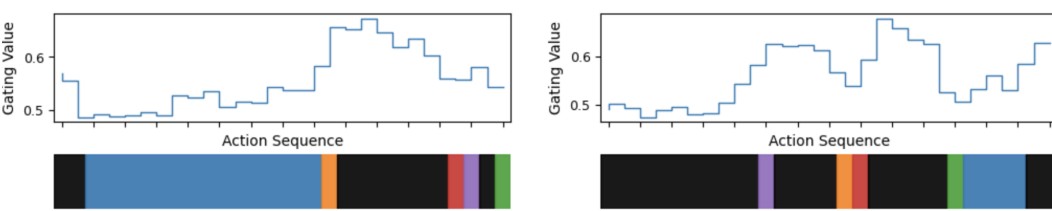

Figure 4: **Visualization of Adaptive Gating Ratio.** The gating values, displayed alongside the action sequence, demonstrate context-aware behavior. In background regions (black), the gate retains historical context; in action regions (colored), it prioritizes current visual evidence.

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

| $f_x$ | $E_Q, E_K$ | $E_V$ | FFN | MLP in Adaptive Gating | MLP in classifier |
|---|---|---|---|---|---|
| LayerNorm | LayerNorm | LayerNorm | Linaer (2048, 1024) | Linear (4096, 512) | LayerNorm |
| ReLU | ReLU | ReLU | GELU | ReLU | ReLU |
| Dropout | Dropout | Dropout | Dropout | Dropout | Dropout |
| Linear ($d_{backbone}$, 2048) | Linear (2048, 512) | Linear (2048, 2048) | Linear (1024, 2048) | Linear (512, 2048) | Linear(2048, $N_c$) |
| ScaleNorm | | | | | |

Table 7: Modeling components in AGA.

| Queue Size | AGA (MT5R) |
|---|---|
| S=4 | 18.4 |
| S=16 | 18.8 |
| S=30 | 18.0 |

Table 8: Ablation study with different queue sizes on EK100.

# A  IMPLEMENTATION DETAILS

For the TSN baseline, we utilized a BN-Inception backbone with pre-extracted features provided by the official EPIC-Kitchens action anticipation repository[1]. For the Swin-B Transformer, we used open-source implementations with pre-trained weights from the *timm* library[2]. Input frames for both architectures were resized to $224 \times 224$ with pixel values rescaled to the range $[-1, 1]$.

The EK100 dataset was sampled at 1 fps over 30-frame sequences, the EK55 dataset at 1 fps over 10-frame sequences, and the EGTEA Gaze+ dataset at 0.5 fps over 10-frame sequences. These sampling configurations align with the anticipation intervals described in the main manuscript.

Figure 7 details the architectures of $f_x$, $E_Q$, $E_K$, $E_V$, the FFN, and the MLPs. We applied a dropout rate of 0.6 on EK100 and 0.4 on both EK55 and EGTEA Gaze+ datasets.

# B  ADDITIONAL EXPERIMENTS

We carried out experiments to determine optimal hyperparameters for AGA and identify its computational cost. With another experiment, we investigated how errors in the predictions $\hat{y}$ impact model accuracy. Finally, we examined how representing prediction uncertainty improves the accuracy of the AGA model.

## B.1  SELECTION OF HYPERPARAMETERS

Two important hyperparameters for AGA are the length $S$ of the queue from which the model can reference past information (Equation 1) and the smoothing factor $\alpha$ in the exponential moving average that determines the temporal window of the query (Equation 2).

### B.1.1  QUEUE LENGTH

The experiments with EK100 reported in the paper were conducted with a queue length of $S = 16$. We also ran experiments for queue lengths 4 and 30. The mean top-5 recall for the different queue lengths is shown in Table 8.

### B.1.2  SMOOTHING FACTOR

Intuitively, EMA injects temporal context into the query and smooths out model prediction jitter. The optimal smoothing constant $\alpha$ depends on factors such as the video sampling rate and the target action duration. As reported in Table 6 of the paper, we found in the EK100 experiment that the MT5R was between 18.4 and 18.8 for $0.2 \leq \alpha \leq 0.8$. This suggests that the accuracy is relatively stable for $\alpha$ in that range.

---

[1] https://github.com/epic-kitchens/C3-Action-Anticipation
[2] https://github.com/huggingface/pytorch-image-models, v0.5.4

| EMA $\alpha$ | Top-1 | Top-5 | MT5R |
|---|---|---|---|
| 0.0 | 15.0 | 34.5 | 15.5 |
| 0.2 | 15.5 | 35.8 | 15.9 |
| 0.4 | 16.1 | **37.4** | 16.4 |
| 0.6 | **16.3** | 37.2 | 16.3 |
| 0.8 | **16.3** | **37.4** | **16.6** |
| 1.0 | 15.7 | 36.1 | 15.9 |

Table 9: Accuracy differences resulting from varying $\alpha$ on EK55.

| EMA $\alpha$ | Top-1 | Recall |
|---|---|---|
| 0.0 | 41.8 | 34.7 |
| 0.2 | 43.9 | 36.2 |
| 0.4 | 44.9 | 37.3 |
| 0.6 | 45.2 | 37.1 |
| 0.8 | **45.4** | **37.4** |
| 1.0 | 43.8 | 36.0 |

Table 10: Accuracy differences resulting from varying $\alpha$ on EGTEA Gaze+.

| Sequence Length | AVT (GFLOPs) | AGA (GFLOPs) |
|---|---|---|
| 8 | 137.28 | 123.91 |
| 16 | 274.56 | 248.29 |
| 32 | 549.12 | 497.57 |

Table 11: Computational cost comparison.

| Sequence Length | AVT (ms) | AGA (ms) |
|---|---|---|
| 8 | 14.97 | 11.4737 |
| 16 | 20.57 | 23.4944 |
| 32 | 33.93 | 47.3397 |

Table 12: Inference time comparison.

We conducted the same study for the other two datasets used in the experiments. Tables 9 and 10 report the results of sweeping the smoothing factor $\alpha$ over $[0, 1]$. We observe that the optimal scores result from a value of $\alpha = 0.8$, which is consistent with the findings on EK100, while even across datasets a choice of $\alpha$ within the range $0.2 \leq \alpha \leq 0.8$ leads to relatively stable accuracy.

## B.2 COMPUTATIONAL COST OF AGA

To evaluate the inference-time computational cost of AGA, we compared it against the AVT baseline. We created random tensors to mimic how sequences of frames are processed in a real inference setting. At each inference iteration we passed the random input to the model and let it do its processing: AVT in one go, due to its transformer architecture, AGA frame by frame, due to its sequential nature.

In Table 11, we show the total number of floating-point operations (FLOPs) computed for both models. AGA required slightly fewer FLOPs, despite using the heavier Swin-B backbone, versus AVT's ViT-B

In Table 12, we show the inference times for AVT and AGA (with backbone Swin-B) measured on a single A100 GPU. The differences can be explained by the way in which the two models process the input. While AGA processes the sequence progressively, AVT handles the entire sequence in parallel.

## B.3 PROPAGATION OF ERRORS

We wanted to understand how robust our model is with respect to errors in the predictions. We set up an experiment where AGA is subjected to inaccurate or noisy predictions that disrupt its action-guidance signal. As input, AGA received 30 frames from a 30-second clip from the EK100 validation set, with 1 second intervals in between. It had to predict the action in the last frame based on the preceding 29. For each sequence, we randomly created a permutation $\pi$ of the 30 numbers $i = 0, \ldots, 29$ and performed 30 runs. In run $i$, the predictions $\hat{y}_{\pi(j)}$, where $0 \leq j < i$, were forcibly reset to a uniform distribution, that is, each action was assigned the probability $\frac{1}{N_c}$. In this manner an increasing random subset of the predictions was made meaningless. In the table below, the first column contains the number of frames for which the the classifier output (i.e., $\hat{y}$) has been reset and the second column shows the resulting mean Top-5 recall (MT5R).

The experiment was run over the entire validation set of EK100. Figure 5 shows that the accuracy steadily decreases from 18.8% to 16.3%.

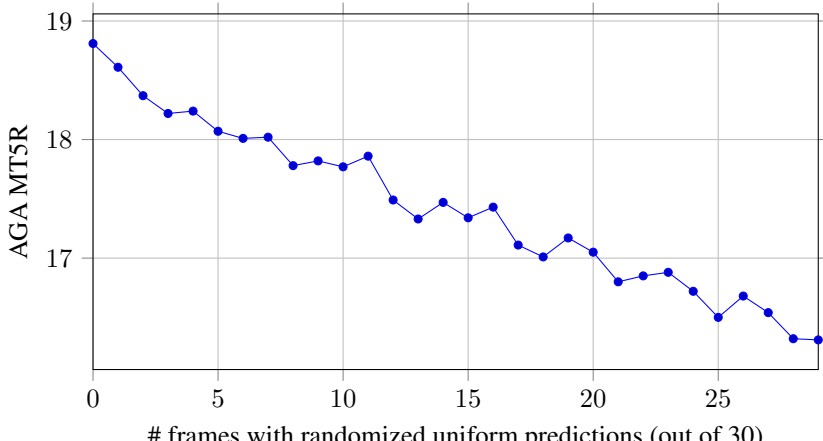

Figure 5: Robustness analysis of AGA against the error occurred in frame prediction.

| Methods | Train On | Inference On | Overall Classes | | | Unseen Classes | | | Tail Classes | | |
|---|---|---|---|---|---|---|---|---|---|---|---|
| | | | Action | Verb | Noun | Action | Verb | Noun | Action | Verb | Noun |
| GT | GT | GT | 16.9 | 30.6 | 36.7 | 14.6 | 34.4 | 26.5 | 16.1 | 25.3 | 32.8 |
| GT | Self-Pred | GT | 17.1 | 31.9 | 36.0 | 14.1 | 32.9 | 26.3 | 16.6 | 26.8 | 32.2 |
| Self-Pred (Top-1 One-Hot) | Self-Pred (Top-1 One-Hot) | Self-Pred (Top-1 One-Hot) | 17.2 | 31.8 | 37.7 | 14.5 | 35.6 | 27.7 | 16.3 | 26.6 | 34.2 |
| Self-Pred (Top-1 One-Hot) | Self-Pred | Self-Pred (Top-1 One-Hot) | 17.5 | 33.4 | 36.4 | 16.3 | 32.6 | 27.7 | 16.5 | 28.4 | 32.5 |
| AGA | Self-Pred | Self-Pred | 18.8 | 32.5 | 38.7 | 16.3 | 34.4 | 28.5 | 18.4 | 27.4 | 35.0 |

Table 13: Comparison between different action-guidance signals for $Q$ and $K$.

## B.4 SELF-PREDICTION VS. GROUND TRUTH ACTIONS

AGA learns predictions $\hat{y}$ that are probability distributions over the set of all action classes with a loss function that measures the difference to a one-hot distributions $y_a$ for a specific action class $a$ as the target.

Intuitively, during its training, AGA strives to generate predictions that are ever closer to the one-hot ground truth. It is therefore natural to ask whether and how the performance of a model changes if the predictions, which are generated to resemble the ground truth, are replaced with actual ground truth distributions. Whether or not this is the case should give us more insight into how AGA works.

We created an experiment where we compared the use of the ground truth as opposed to the self-predicted $\hat{y}$ in query and keys. For frames with an action label $a$, we represented the ground truth with a one-hot $y_a$. For frames without label, occurring in background our transition segments, we used one-hot vectors representing a generic background class.

For additional insight, we included experiments where we simplified the self-predicted distributions to one-hot vectors $\hat{y}_{\top 1}$, carrying a 1 for the action with the highest probability in $\hat{y}$, thus indicating the Top-1 predicted action, and in this way aligning the representation to the ground truth format.

For both formats, ground truth and one-hot self-prediction, we ran two versions of the experiment: one where we used the revised format for both training and inference, and another one where we used it only for the inference.

We ran the experiments on the EK100 validation set. The results are shown in the Table 13. Here, "GT" stands for Ground Truth, that is, usage of the one-hot $y_a$, "Self-Pred" for Self-Predication, that is, usage of the $\hat{y}$, and "Self-Pred (Top-1 One-Hot)" for the usage of the vectors $\hat{y}_{\top 1}$. The figures refer to the accuracy on EK100 and the columns are identical to those in Tables 1 and 2 of the paper.

The results show that AGA, which leverages the full self-prediction distribution, achieves the highest accuracy. This can be intuitively explained by three factors:

- The full self-prediction carries richer intrinsic information than a one-hot ground truth label (as further evidenced by the accuracy drop when binarizing the self-prediction into one-hot);

- Self-prediction still provides meaningful signals when no action is annotated during the observation window (for instance, background or transition segments);

- Although GT provides the correct Top-1 future label, it lacks information about alternative plausible futures, whereas the (imperfect but semantically informative) self-prediction distribution better reflects the uncertainty and structure of the observations.

## C  MORE QUALITATIVE EXAMPLES

We provide additional qualitative examples from the EK100, EK55, and EGTEA Gaze+ datasets in Figures 6-11, where clips (a)–(d) illustrate successful predictions, with the ground-truth action revealed at the latest timestep, and clips (e)–(h) highlight failure cases in which the top-5 prediction of the model omit the target action. Figures are best viewed at full width and zoom in for full detail.

### C.1  EPIC-KITCHENS-100 (EK100)

Figure 6 presents successful cases from the EK100 dataset. In video **(a)**, the scene evolves slowly and the washing activity is clear. The noun *plate* is also unambiguous, which allows the model to converge on *wash plate*, while alternative verbs such as *hold*, *insert*, and *put* are gradually ruled out. In video **(b)**, the target action *throw bin* occurs between the third and seventh frames and is correctly identified by the model. Video **(c)** shows a case with little movement, where the verb prediction narrows to a small set of consistent options over time, and the model ultimately selects the correct object. Video **(d)** is more challenging because the subject intent is unclear. The model nonetheless predicts a coherent sequence: *throw food* followed by *close bin*, while assigning lower probability to *open bin* once the hand interaction with the bin becomes visible.

Figure 7 illustrates failure cases. In video **(e)**, the verb is predicted correctly but the noun is misclassified as *pizza*. In videos **(f)**–**(h)**, errors result either from insufficient visual detail (**(f)**) or from ambiguity introduced by multiple plausible object candidates (**(g)** and **(h)**).

The EK100 examples demonstrate that the model can handle everyday kitchen activities with gradual scene changes, but performance degrades when object categories are visually similar or when frames lack sufficient detail.

### C.2  EPIC-KITCHENS-55 (EK55)

Figure 8 highlights successful cases from the EK55 dataset, including challenging scenarios. In videos **(a)** and **(b)**, the target objects (*container* and *board:cutting*, respectively) are not visible in the frames. Nevertheless, the model anticipates the actions *take container* and *take board:cutting* by leveraging contextual cues. Video **(c)** demonstrates the capacity to distinguish between similar actions, predicting *put plate* rather than *take plate*, based on the subject intention inferred from earlier frames. Accurate prediction requires retention of long-term evidence, which is supported by the adaptive gating mechanism and the AGA design. In video **(d)**, the ongoing activity of cooking pasta is anticipated, with the model predicting *put-down spoon* as the next action in the final frame.

Figure 9 presents failure cases where the model encounters ambiguity or insufficient context. In video **(e)**, the object *tofu* is occluded, leading to a misprediction as *container* because the opaque surface prevents visibility of the contents. Video **(f)** illustrates noun misclassification caused by visual similarity, with *olive* confused for *celery*. In video **(g)**, the verb is misclassified because the hand interaction with *oil* is only partially visible. In video **(h)**, the model confuses *chilli* with visually similar objects such as *tomato* and *pepper*.

The EK55 examples highlight the advantage of AGA and adaptive gating for leveraging long-term context, while also revealing difficulties when objects are occluded or visually confusable.

### C.3 EGTEA Gaze+

Figure 10 provides qualitative examples from the EGTEA Gaze+ dataset, covering both successful and challenging scenarios. In video **(a)**, the model captures subtle visual cues, such as the sponge disappearing from the bottom edge of the frame, which leads to the correct prediction of *put sponge*. In video **(b)**, the model predicts *put container* and links the container to the *tomato* by recalling information from the initial frames, demonstrating the use of long-term memory to form object–action associations. Video **(c)** shows a case where observed movement suggests the verb *take* instead of the ground-truth verb *put* for the *bread:container*. Although direct evidence for returning the bread to the refrigerator is absent, the correct action still appears as the second-ranked prediction, reflecting the model strategy of coupling related verbs such as *put* and *take*. In video **(d)**, the model addresses a sequential scenario involving *take cheese:container*, where one action is visible and the other remains hidden. Accurate recognition requires identifying the container holding the cheese, which can only be inferred from the presence of its cover.

Figure 11 presents failure cases caused by insufficient evidence or excessive ambiguity. In videos **(e)** and **(f)**, the absence of salient details leads to incorrect predictions. Video **(g)** demonstrates noun ambiguity: the model predicts verbs confidently but assigns nearly equal probability to multiple visible objects. In video **(h)**, the observation provides minimal context, leaving the model unable to determine the correct action.

The EGTEA Gaze+ examples highlight the model capacity to connect subtle visual cues with long-term memory, while also revealing the difficulty of disambiguating actions in situations with limited observations or an excessive number of potential object candidates.

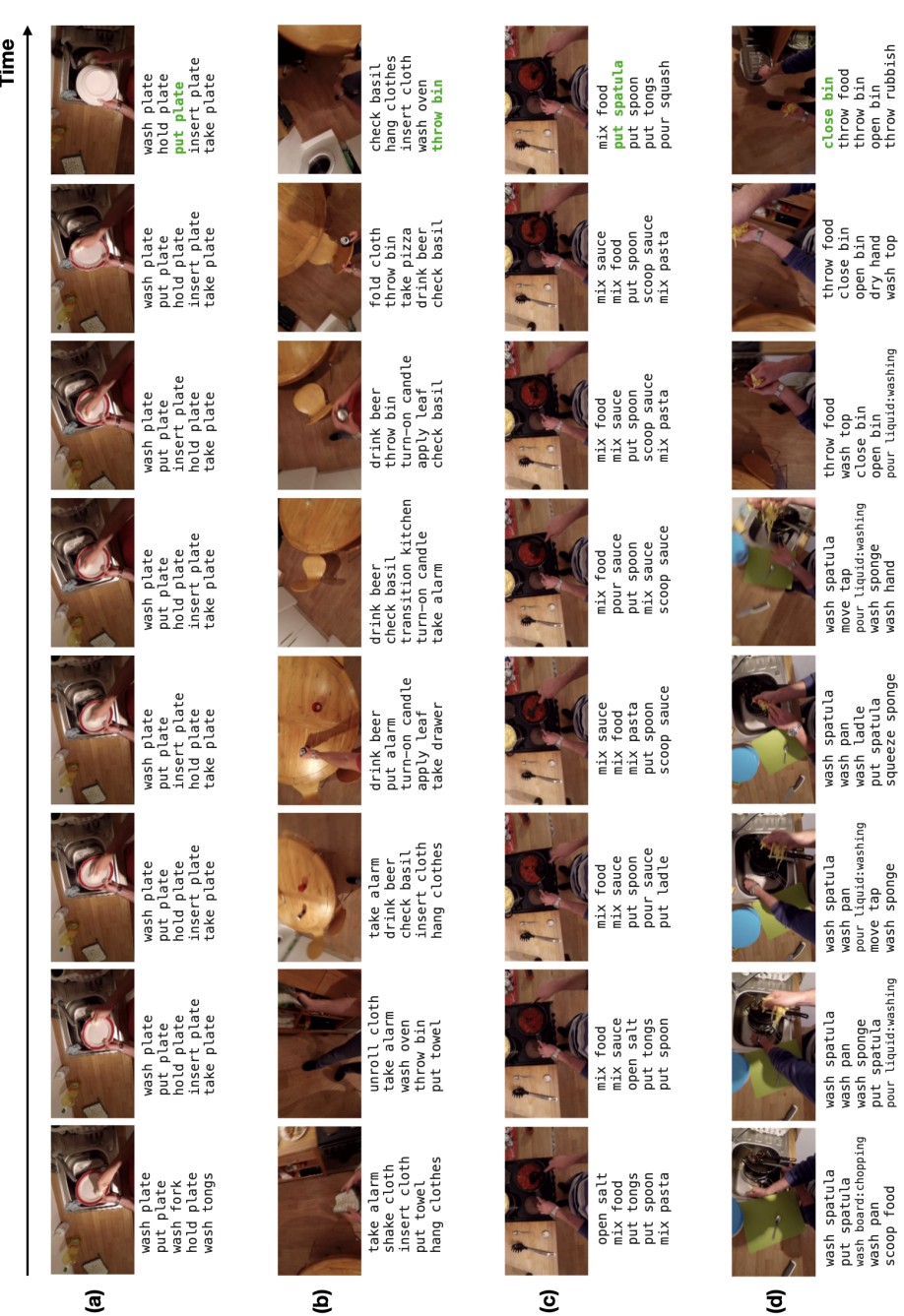

Figure 6: Four video clips from the **EPIC-Kitchens-100** validation set, illustrating **correct** predictions. Each clip demonstrates the last eight frames along with the top-5 action anticipations at $\tau_a = 1s$, with the ground truth highlighted in bold green.

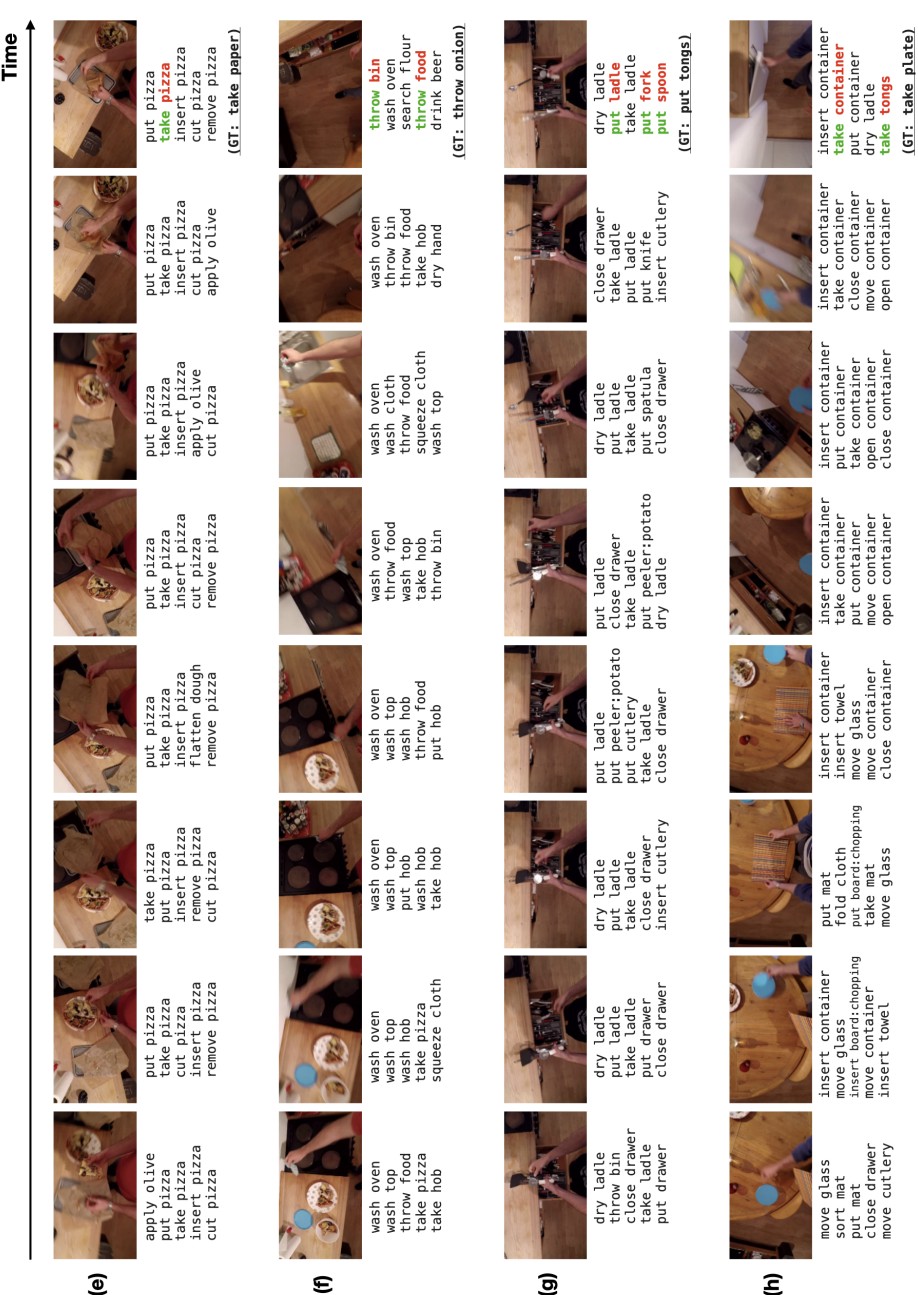

Figure 7: Four video clips from the **EPIC-Kitchens-100** validation set, showcasing **incorrect** predictions. Each clip demonstrates the last eight frames and the top-5 action anticipations at $\tau_a = 1s$, with the ground truth revealed in the final frame. The correct verb and noun are highlighted in bold green.

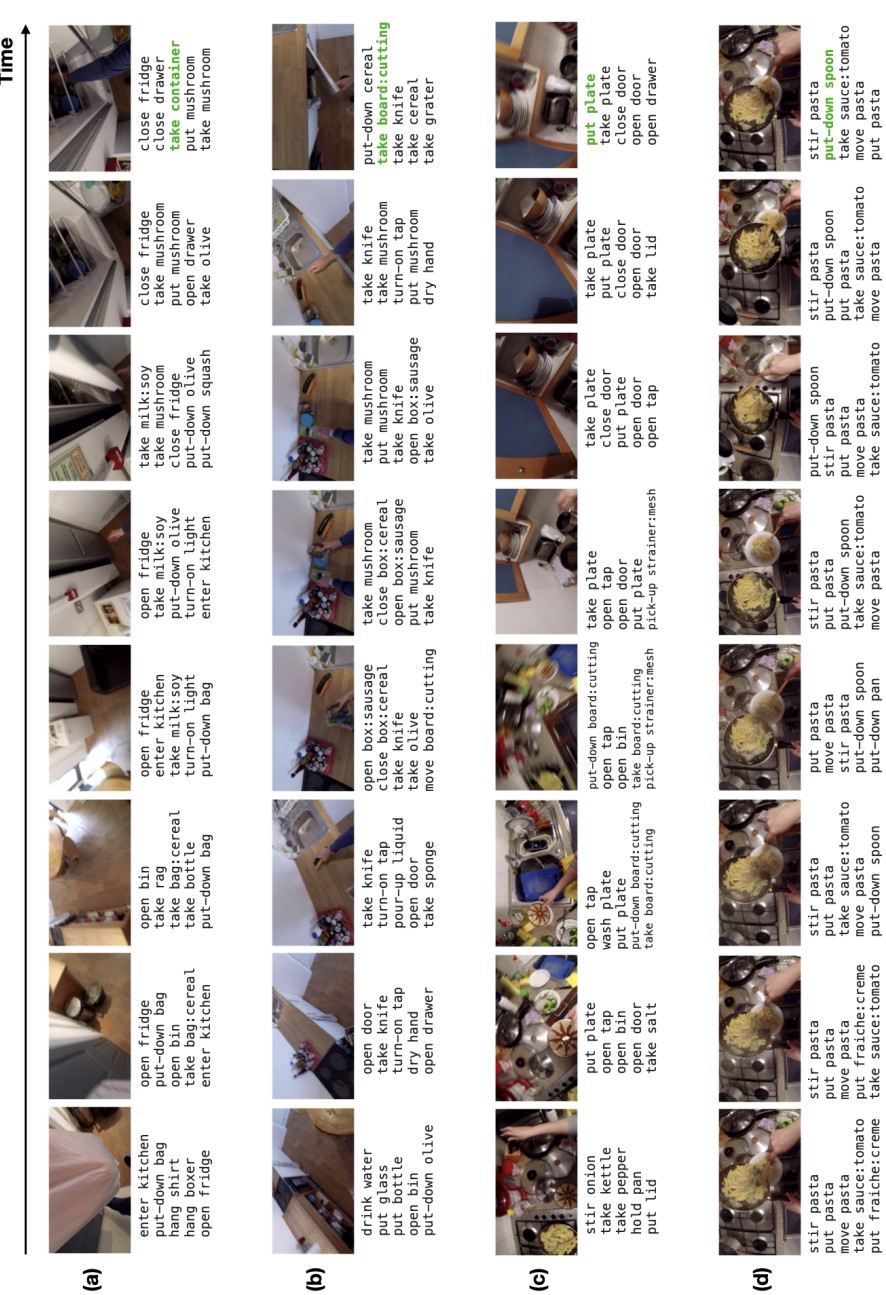

Figure 8: Four video clips from the **EPIC-Kitchens-55** validation set, illustrating **correct** predictions. Each clip displays the last eight frames and the top-5 action predictions. The ground truth is highlighted in bold green in the latest frame, occurring at $\tau_a = 1s$.

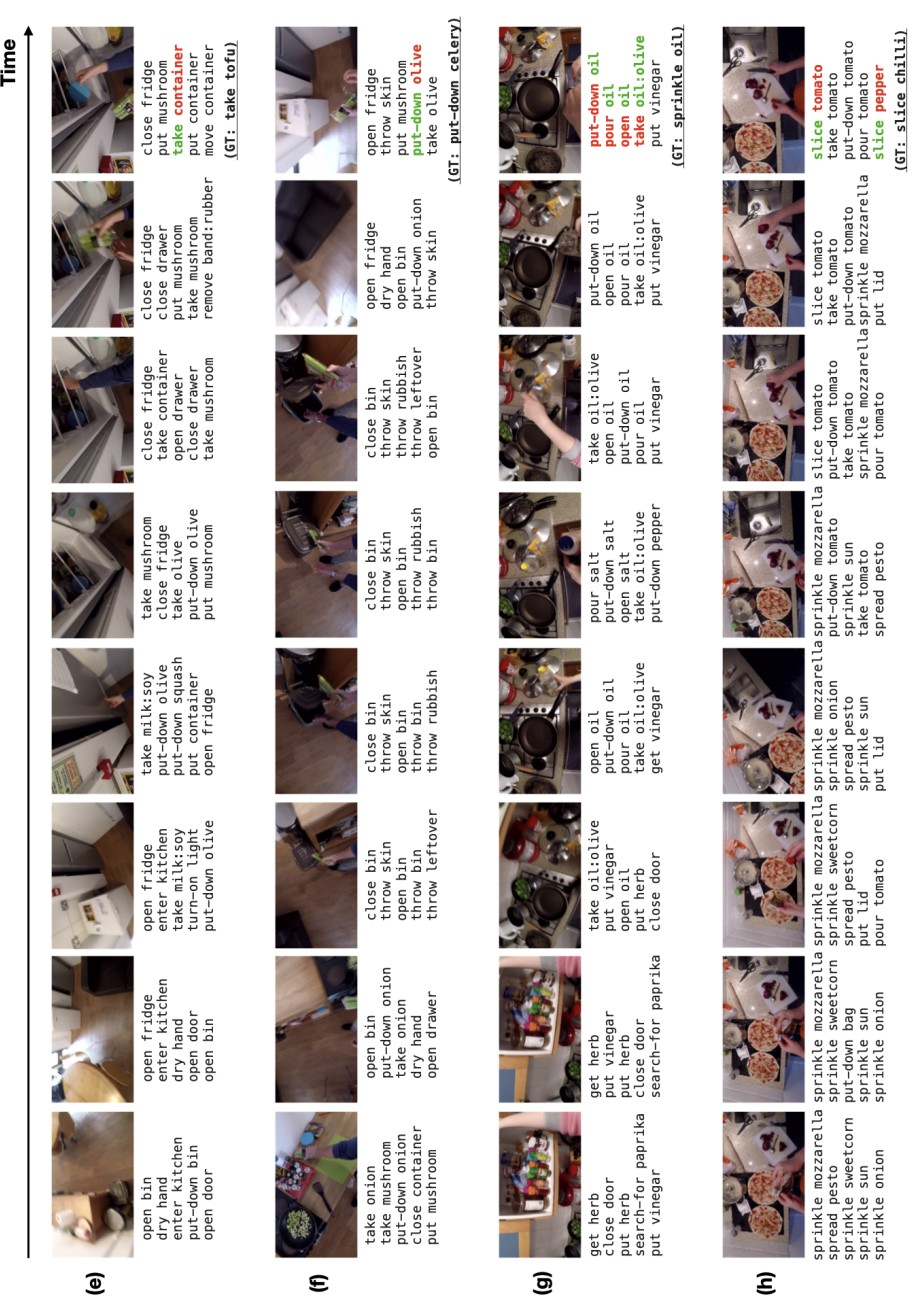

Figure 9: Four video clips from the **EPIC-Kitchens-55** validation set, showcasing **incorrect** predictions. Each clip displays the last eight frames and the top-5 action predictions. The ground truth is highlighted in bold green in the latest frame, occurring at $\tau_a = 1s$.

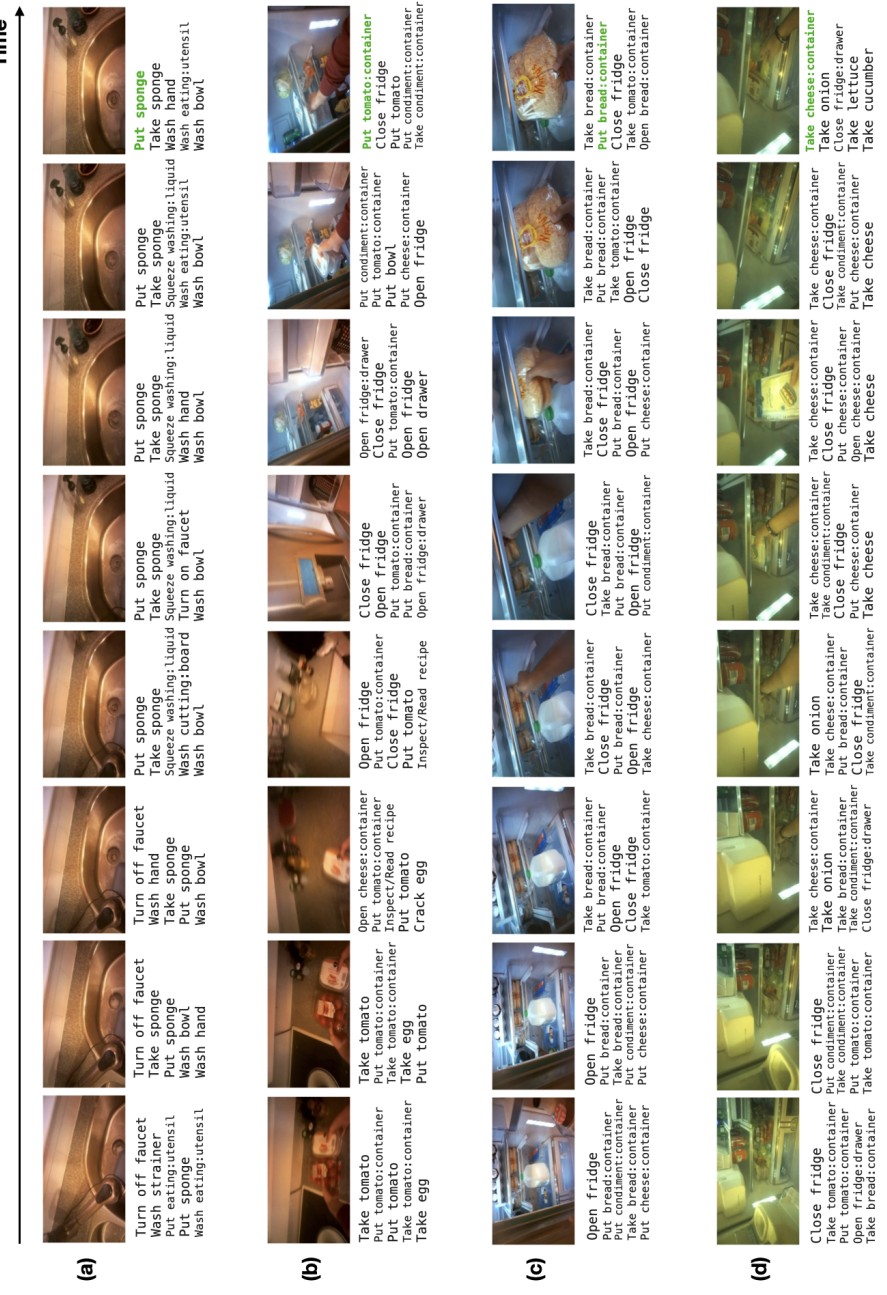

Figure 10: Four video clips from the **EGTEA Gaze+** validation set, highlighting **correct** predictions. Each clip displays the last eight frames and the top-5 action predictions. The ground truth is highlighted in bold green in the latest frame, occurring at $\tau_a = 0.5s$.

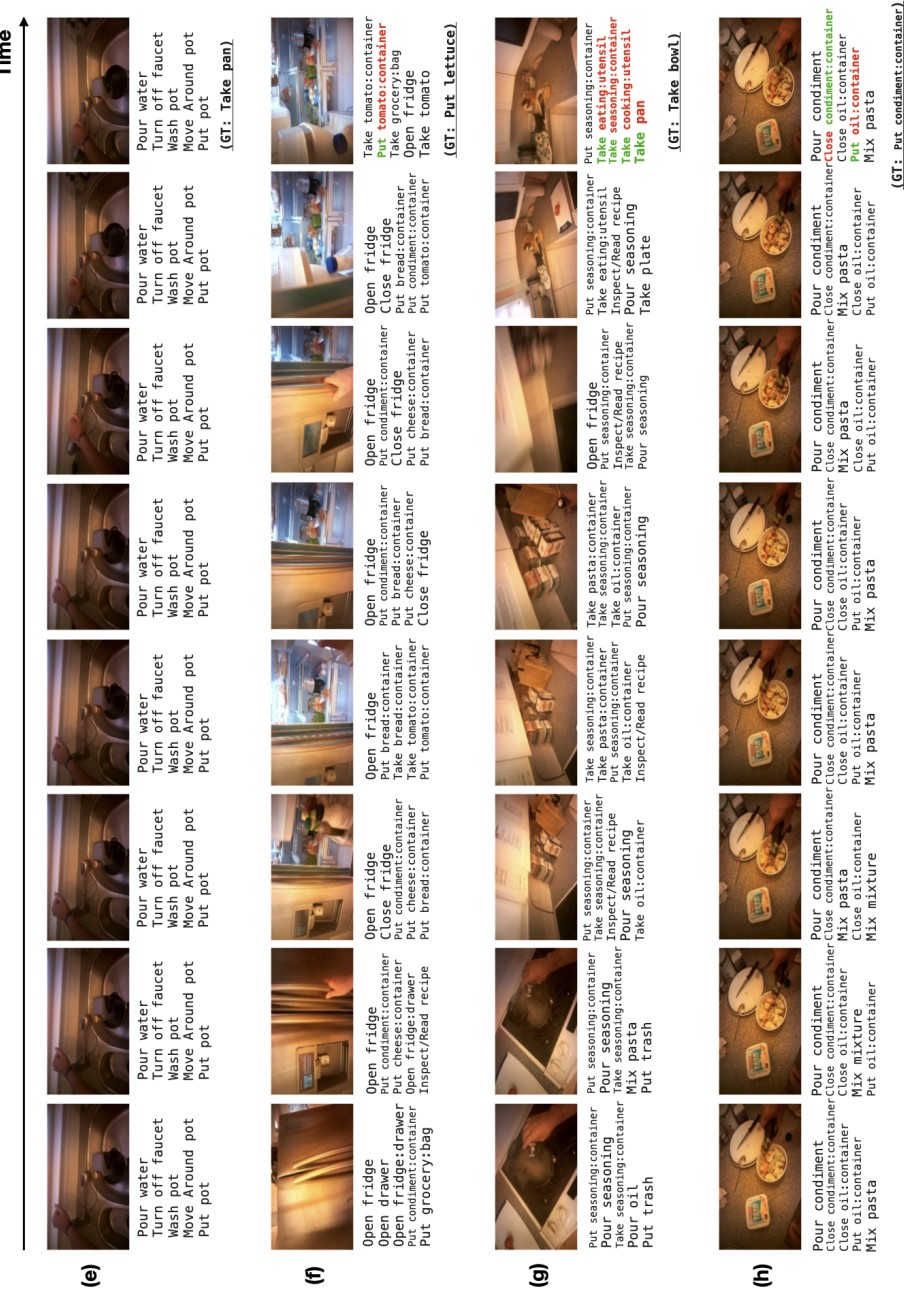

Figure 11: Four video clips from the **EGTEA Gaze+** validation set, showcasing **incorrect** predictions. Each clip displays the last eight frames and the top-5 action predictions. The ground truth is highlighted in bold green in the latest frame, occurring at $\tau_a = 0.5s$.

## D    LLMS USAGE

The authors utilized LLM powered AI tools (GPT-5 and Grammarly) to proofread sentences and identify grammatical errors.

