# OpenReview forum: "Action-Guided Attention for Video Action Anticipation"
_ICLR.cc/2026/Conference — ICLR 2026 Poster_

### Official Review · Reviewer_TEVQ · 2025-10-27

**Soundness:** 3
**Presentation:** 2
**Contribution:** 3
**Rating:** 6
**Confidence:** 4

**Summary:**

This paper focuses on the task of long-term activity anticipation in videos. The main technical contribution is the advocacy of using model frame-wise raw output probability as feature / input to another attention process, where the self-produced predictions serve both as query and key, meanwhile the frame embedding as the value. Moreover, standard attention for long-range (temporal) modelling is identified as sub-optimal in this paper, and authors instead propose to use standard temporal smoothing, i.e. EMA and adaptive filtering, i.e., gating, techniques in the model for better performance. Strong experimental results are demonstrated to support the method.

**Strengths:**

A. The writing is smooth, math easy to read and overall clarity is good. The experimential results are strong across 3 different datasets for activity anticipation, which demonstrate the generalization ability of the proposed approach. The overall approach is quite simple and easy to implement.

B. This reviewer has found the AGA design interesting, though less well-structured / supported throughout the documents. The way AGA works is: Action prediction (distribution over all predefined classes), \hat{y}, is the self-produced outputs and it then works on the incoming frame-level embedding (pre-extraced) progressively along the time axis for next-step prediction. This core design bears a lot similarity to the autoregressive cross-attention~ish fashion, which, however, differs most critically in the information abstraction level - the AGA module works on the semantic level abstraction, e,g., re-use the raw probability prediction. A summary of Q/K/V design against the orignal applications (transformers) is as follows.

| Model                                  | Query Source             | Key/Value Source                                       | Category                                       |
| -------------------------------------- | ------------------------ | ------------------------------------------------------ | ---------------------------------------------- |
| **Transformer Encoder**                | Same input sequence      | Same sequence                                          | Self-attention                                 |
| **Transformer Decoder (e.g., in GPT)** | Generated tokens         | Encoder outputs                                        | Cross-attention                                |
| **AGA**                                | EMA of predicted actions | Past actions (semantic) / past frame features (visual) | **Semantic-Visual Cross-Attention** |

The intution of transformer, by the best knowlege of this reviewer, is that re-weighting the “Value” vectors using relationships between “Query” and “Key” vectors. In this case, AGA is re-weighing the past frame visual representations using the relationship between semantic query and key vectors, which fits the demand of this paper. However, this leads to the 1st weakness point in the next section.

**Weaknesses:**

A.  It remains unclear what exactly make the AGA working for this task, and hinder readers summarize useful take-away message from this paper. Here are some questions to start the discussion.

- According to the statement, quote "only rely on dot-product attention over pixel representations...", it seems the key finding here is that information in visual tokens being too preliminary, and the semantic feature is better. Yet, isn't \hat{y} derived out of the pixel representation, e, in the first place through a output head, f_{\theta}? So, what is the magic here empowering \hat{y} with stronger information after some MLP projection?  Or say it in another way: if the pixel representations already encode the necessary semantics, why does the paper inject predicted logits (the model’s own action scores) into attention — instead of operating on the hidden-state space directly? A quick extra experiment using last-layer hidden states might shed lights on this matter.

- The paper argues that visual cues might make the model prone to over-fitting. This reviewer would doubt if that's still true after the second encoder, f_{x}, especially after a few epochs of training. This argument is never supported with experimential study. A good one could be monitoring the validation set loss under two different settings - one with semantic feature and another one with visual features along training epochs.

- The cross-attention alike mechansim in AGA seems unpredecent from previsouly published work. AVT (Anticipative Visual Transformer) can be seen as the canonical autoregressive transformer baseline, which only tested self-attention on visual tokens, not the cross-attention over past visual features and the next incoming frame-level feature. Maybe a study of a direct comparison where Q = current visual embedding and K/V = past visual embeddings (true autoregressive cross-attention)  would isolate the benefit of using semantic (action) representations from the benefit of directional cross-attention.

- The EMA and gating techniques can be implicitly achieved using attention. Prior-driven technique are powerful but less generic. It seems the main motivation here is resolving the training difficulity. Replacing generic module with prior-driven one does not seem to be a scalable method. Maybe the authors can share the failure logs (training logs) of using learnable components (such as pure attention). The community might come up better optimization idea for this issue.

B. The training recipe is missing quite a few important information for re-production purpose:
- frame sampling rate
- queue initiation - It’s unclear how the queue is filled at the start of a clip (zeros? repeats? skip first S frames?).
- Any data augmentation?
- EMA update scope - Is EMA applied to the raw logits or the softmax output (normalized prob.)?
- Again on the past prediction, y_{t-1: t-s}, are they also being optimized with the cross-entropy loss? If so and they are produced auto-regressively, are the BTPP algorithm used to optimized a sequence of inter-dependent predictions or separately?
- Prediction head architecture - the exact dim and layers of MLP here.


C. The post-training analysis study is a bit pre-mature. The Forward and Backward analysis are interesting and visually compelling but lack methodological transparency and quantitative validation.
Key implementation details (which layer, normalization, gradient target) are omitted, and no formal evaluation is conducted to confirm that these analyses genuinely capture causal or influential dependencies. This reviewer would not think of it as a well-rounded interpretability study.

**Questions:**

A. How is the past predictions, \hat{y}_{t-1: t-s}, produced? The reviewer guess that during training, the model is unrolled over S steps and computes each \hat{y} auto-regressively, feeding those predictions into the next step’s queue. Please confirm the details.

B. Add an equation for producing \hat{y}.

C. This reviewer believe the following work on acitivity anticipation are revalent for discussion.
- When will you do what?- Anticipating Temporal Occurrences of Activities, CVPR 2018,
- Time-Conditioned Action Anticipation in One Shot, CVPR 2019,
- On diverse asynchronous activity anticipation, ECCV 2020,

---

> ### Author Response · Authors · 2025-11-25
> **Response to Reviewer TEVQ (Part 1)**
>
> We respond to the questions, weaknesses and comments raised by the reviewer:
>
> $${}$$
>
> **Question 1**
> > A. How are the past predictions, $\hat{y}_{t-1: t-s}$, produced? The reviewer guesses that during training, the model is unrolled over S steps and computes each $\hat{y}$ auto-regressively, feeding those predictions into the next step’s queue. Please confirm the details.
>
> At the first frame, when the queue is empty, the attention module is bypassed and a zero vector is pushed to the queue. Starting from the second frame, the queue contains at least one timestep for both Q and K, allowing the attention mechanism to begin processing information even if the queue is not yet full. We will clarify this in the paper.
>
> $${}$$
>
> **Question 2**
> > Add an equation for producing $\hat{y}$.
>
> At the end of section 3.2 we will add the equation
> $$
>     \hat{y}_t = \text{softmax}(\text{MLP}(o_t)),
> $$
> where $o_t$ is the output of the adaptive gating component. Details of the MLP design to map $o_t$ to our prediction $\hat{y}_t$ were already provided in Table 7 of the Supplementary Material.
>
> $${}$$
>
> **Question 3**
> > This reviewer believes the following papers on activity anticipation are relevant for discussion.
> When will you do what? - Anticipating Temporal Occurrences of Activities, CVPR 2018,
> Time-Conditioned Action Anticipation in One Shot, CVPR 2019,
> On diverse asynchronous activity anticipation, ECCV 2020,
>
> We agree that these papers are relevant and we will refer to them in the first paragraph of Section 2, Related Work. Here we furthermore contrast their contributions to those of our work:
>
> Farha et al. (When will you do what? - Anticipating Temporal Occurrences of Activities, 2018) also argue that anticipating in the semantic label space is more effective than directly forecasting from pixels, which is consistent with our motivation. However, their RNN/CNN predictors never feed predictions to interact with the visual encoder features and are solely used to generate the prediction in the action space. In contrast, AGA injects the model’s own action distribution into the attention mechanism as queries and keys, so that future action semantics directly guide where the visual attends.
>
> Ke et al. (Time-Conditioned Anticipation, 2019) refine their anticipation prediction by conditioning on the injected temporal features with an attention mechanism. In contrast, AGA uses the model’s own future action distribution as the query and key of the attention mechanism.
>
> Zhao and Wildes (On Diverse Asynchronous Activity Anticipation, 2020) model future activities by conditioning the initial observation in the discrete action–time sequence through a conditional GAN to obtain diverse multi-modal futures; their predictions are never used to guide visual feature extraction. AGA is, on the other hand, an attention module that feeds the model’s own action scores back into the attention mechanism to reweight the visual representation.
>
> $${}$$
>
> **Weakness 1**
> > According to the statement, quote "only rely on dot-product attention over pixel representations...", it seems the key finding here is that information in visual tokens being too preliminary, and the semantic feature is better. Yet, isn't $\hat{y}$ derived out of the pixel representation, $e$, in the first place through an output head, $f_{\theta}$? So, what is the magic here empowering $\hat{y}$ with stronger information after some MLP projection? Or say it in another way: if the pixel representations already encode the necessary semantics, why does the paper inject predicted logits (the model’s own action scores) into attention — instead of operating on the hidden-state space directly? A quick extra experiment using last-layer hidden states might shed lights on this matter.
>
> In AGA, there is no recurrent hidden state: the model processes each frame independently, and the propagated signal comes from the predicted action distribution. Consequently, the attention mechanism is intentionally designed so that the model can reason over actions rather than raw pixel-level embeddings. This is not an emergent effect but a structural constraint we impose. The performance gain is not attributed to MLP projections, but from the architectural constraint: the attention module is restricted to operate on the semantic space defined by action predictions. This enforces an action-centric reasoning process by design.
>
> $${}$$
>
> For further responses see Response to Reviewer TEVQ (Parts 2 and 3)

---

> > ### Author Response · Authors · 2025-11-25
> > **Response to Reviewer TEVQ (Part 2)**
> >
> > **Weakness 2**
> > > The paper argues that visual cues might make the model prone to over-fitting. This reviewer would doubt if that's still true after the second encoder, $f_{x}$, especially after a few epochs of training. This argument is never supported with experimental study. A good one could be monitoring the validation set loss under
> > two different settings - one with semantic feature and another one with visual features along training epochs.
> >
> > According to Table 5 and our response to Weakness 3, using frame embeddings cannot match the performance obtained with action-guided queries and keys. This is because the action-guided formulation constrains the attention mechanism to operate in the action space, rather than relying on visual observations that may capture coincidental or irrelevant cues and nuances in the frames. In this context, we use the term overfitting to describe the model’s tendency to attend to such clutter or spurious correlations, which would not generalize to unseen samples. By contrast, the action-guided semantics provide a more stable and task-aligned representation, reducing the likelihood of such misfocused attention patterns.
> >
> > $${}$$
> >
> > **Weakness 3**
> > > The cross-attention alike mechanism in AGA seems unprecedented from previously published work. AVT (Anticipative Visual Transformer) can be seen as the canonical autoregressive transformer baseline, which only tested self-attention on visual tokens, not the cross-attention over past visual features and the next incoming frame-level feature. Maybe a study of a direct comparison where Q = current visual embedding and K/V = past visual embeddings (true autoregressive cross-attention) would isolate the benefit of using semantic (action) representations from the benefit of directional cross-attention.
> >
> > We conducted the requested experiment which suggests an advantage of AGA over auto-regressive cross-attention:
> >
> > | Queue Size | Q=current visual embedding; K/V=past visual embedding | AGA  |
> > |------------|--------------------------|-----------------------|
> > | S=4        | 17.4                     | **18.4**              |
> > | S=16       | 17.7                     | **18.8**              |
> > | S=30       | 17.4                     | **18.0**              |
> >
> > $${}$$
> >
> > **Weakness 4**
> > > The EMA and gating techniques can be implicitly achieved using attention. Prior-driven techniques are powerful but less generic. It seems the main motivation here is resolving the training difficulty. Replacing a generic module with a prior-driven one does not seem to be a scalable method. Maybe the authors can share the failure logs (training logs) of using learnable components (such as pure attention). The community might come up with better optimization ideas for this issue.
> >
> > We thank the reviewer for the suggestion and agree that future work may uncover more general or learnable alternatives. We want to clarify that although EMA and gating inject lightweight priors, they are empirically effective and consistently stable on the datasets studied. Scalability is not the primary goal of this work: we propose an alternative attention design that specifically takes the action semantics into consideration.
> >
> > $${}$$
> >
> > **Weakness 5**
> > > The training recipe is missing quite a few important informations for re-production purposes:
> > > 1. Frame sampling rate
> > > 1. Queue initiation - It’s unclear how the queue is filled at the start of a clip (zeros? repeats? skip first S frames?).
> > > 1. Any data augmentation?
> > > 1. EMA update scope - Is EMA applied to the raw logits or the softmax output (normalized prob.)?
> > > 1. Again on the past predictions, $y_{t-1: t-s}$, are they also being optimized with the cross-entropy loss? If so and they are produced auto-regressively, is the BTPP algorithm used to optimize a sequence of inter-dependent predictions or separately?
> > > 1. Prediction head architecture - the exact dim and layers of MLP here.
> >
> > The missing details are as below:
> > 1. Frame sampling rate: 1 FPS for EPIC-Kitchens-55/100; and 2 FPS for EGTEA Gaze+.
> > 1. Queue initiation: Please refer to the response to Question 1, where we have explained this.
> > 1. Data augmentation: We don’t apply any augmentation in this study.
> > 1. EMA update scope: The EMA is applied to the softmax probability.
> > 1. Optimization of past predictions: Yes, past predictions are optimized with the cross-entropy loss if the dataset includes annotations for the observation duration (such as in EPIC-Kitchens). EGTEA Gaze+, however, lacks dense annotations. In that case, $\hat{y}_{t-1:t-S}$ receives no supervision signals. The training gradient always spans the entire sequence, since each prediction depends on the attention output, which is a weighted sum of all previous frames.
> > 1. Prediction head architecture: In Table 7 in the Supplementary Materials we have listed the set-up of all components of AGA, including the size of the different layers.

---

> > > ### Author Response · Authors · 2025-11-25
> > > **Response to Reviewer TEVQ (Part 3)**
> > >
> > > **Weakness 6**
> > > > The post-training analysis study is a bit pre-mature. The Forward and Backward analysis are interesting and visually compelling but lack methodological transparency and quantitative validation. Key implementation details (which layer, normalization, gradient target) are omitted, and no formal evaluation is conducted to confirm that these analyses genuinely capture causal or influential dependencies. This reviewer would not think of it as a well-rounded interpretability study.
> > >
> > > We appreciate the reviewer’s thoughtful feedback. Our forward and backward analyses are designed in the same spirit as gradient-based interpretability methods such as GradCAM, where they aim to qualitatively visualize the evidence and patterns captured by AGA, rather than serve as a full causal analysis framework. Furthermore, existing annotations on datasets of interest do not support such a formal evaluation.

---

> > ### Comment · Reviewer_TEVQ · 2025-11-25
> >
> > > reason over actions rather than raw pixel-level embeddings
> >
> > Could the authors confirm / correct: Are actions coming from a projection (e.g., MLP) from pixel-level embeddings?

---

> > ### Comment · Reviewer_TEVQ · 2025-11-25
> >
> > > ... the architectural constraint ...
> >
> > Thanks. It is clear to this reviewer that this paper praises action-level attention over pixel representation attention, which is doable and acceptable as long as well motivated and theoretically well supported. What has been demonstrated in the main submission and in **the response to weakness 2** are evidences / testimony to the advertised assumption.
> >
> > Yet, this reviewer seeks deeper understanding. There are many layers between the introduction of action-level cross attention and the final improved results.
> >
> > If tracing back to the original input pixel domain is not possible, monitoring logs over the training/val/eval processes can also shed some lights - after all the entire approach is a machine learning system in nature. The action-level attention ought to bring some changes to the learning process so that a better generalization ability or lower loss plateau is achieved. Such an observation in contrast to that of using the pixel-representation is the most critical take-away message to general audience.
> >
> > Alternatively the authors can explain what is the definition of **richer intrinsic information** mentioned in the general response. Is it just a soft version of one-hot label?
> >
> > Overall, this reviewer would go against overlooking root cause and justify idea with unclear terminology.

---

> > > ### Author Response · Authors · 2025-12-02
> > > **Response to Reviewer TEVQ: Richer intrinsic information**
> > >
> > > We refer to “richer intrinsic information” as the underlying action-transition dynamics that unfold over time. This goes beyond simply providing a softer version of the one-hot GT label. Specifically: (1) GT denotes only a single future outcome, ignoring alternative plausible futures; (2) GT provides no information about how an action evolves within its duration; and (3) GT is absent during background or transition periods. In contrast, the model’s self-prediction captures how action likelihoods change over time and thus reflects, to some degree, these intrinsic temporal dynamics.

---

> ### Author Response · Authors · 2025-12-02
> **Response to Reviewer TEVQ: Actions coming from a projection?**
>
> Yes, AVA uses MLP projections and affine transformations as basic building blocks with trainable parameters. It combines MLP and affine projections as basic compute nodes into a layered computation graph that outputs target action probabilities. Note that this holds for most deep learning methods and inference tasks including GPT style Transformers, ViT, Swin Transformer, AVT and all action anticipation models in Tables 1, 2, 3, 4 and 5 of the main paper. Methods differ in the way these basic building blocks are combined: these are the model design choices and most research is about model architecture design. At the very core, even LSTM and Transfromer differ just in the way such basic MLP projection and affine transform compute nodes are combined – with forget,reset,update,output gates the former, and multi-head self-attention the latter.
>
> The model for action anticipation proposed in this paper combines MLP and affine projections in a novel, autoregressive style manner with cross attention and gating mechanism. It would therefore not be correct to narrow down our contribution to “actions coming from projections (e.g. MLP) from pixel level embeddings”:
>
> While the last computation to produce action probabilities is an MLP projection, the input to it is a convex combination of frame-wise embedding $e_t$ and the historical context $\tilde{h_t}$: $\tilde{h}_t$ is obtained with our instantiation of semantic-visual cross-attention, as indicated by this reviewer in the table (section Strengths, B.); $e_t$ is the frame-level semantic feature representation of pixel-level evidence extracted by a backbone (Swin-B in our experiments); the convex combination is realised through self-gating, via $g_t(e_t, \tilde{h_t})$.
>
> Through self-gating, AGA can learn to ignore frame-level features $e_t$, ignore historical context $\tilde{h_t}$ , or mix them to produce action probabilities $y_t$. It does so at each time step and feature-wise: at inference, it can ignore features of $e_t$ when computing $y_t$ while reusing them for $y_{t+2}$ for example.
>
> Through cross-attention, AGA learns to aggregate and encode past $S$ frame-level features into the historical context $\tilde{h}_t$ to obtain $y_t$. It uses dot-product attention with MLP projections of past action probabilities as key and queries to aggregate past frame-level features and inference $y_t$.
>
> Keys and query in cross-attention are computed (through MLP projections and exponential moving averaged MLP projections) from past model outputs, resulting in a autoregressive style update of model outputs. More precisely, model outputs are computed from the last $S$ frame-level features, however, the attention weights applied on these depend on all previous outputs and thus, capture long-term dependencies. As a consequence, during training all MLP projections involved in computing keys and query receive back-propagated gradient contributions from all frame embeddings, not just from the last $S$. The parameters of these MLP projections are shared across time, hence boosting data-efficiency in training.
>
> The action probabilities are therefore inferenced from a sequence of frame-level features through a layered autoregressive-style computation with cross-attention and gating mechanism, resulting in a dense computation graph with NLP projections and affine transformations as basic compute nodes.

---

### Official Review · Reviewer_XXT2 · 2025-10-31

**Soundness:** 3
**Presentation:** 3
**Contribution:** 3
**Rating:** 6
**Confidence:** 4

**Summary:**

The manuscript proposes Action-Guided Attention (AGA), a semantic attention mechanism for video action anticipation. Instead of relying on pixel-level similarity, AGA uses predicted action distributions as queries and keys to focus on temporally relevant past frames. An adaptive gating module further balances historical and current visual cues. The authors also introduce forward and backward analyses to interpret the model’s learned dependencies. Extensive experiments on EPIC-Kitchens-100, EPIC-Kitchens-55, and EGTEA Gaze+ demonstrate consistent improvements over prior methods.

**Strengths:**

1. The manuscript proposes a novel approach that leverages the model’s own predicted action distributions to guide attention, enabling it to focus on high-level contextual cues rather than low-level visual similarity. This design offers a fresh and effective perspective for modeling in action anticipation.
2. The method shows consistent improvement on tail and unseen classes, indicating enhanced robustness across class distributions compared to prior work.
3. The proposed forward and backward analyses offer interpretable insights into how the model captures causal dependencies in its high-level representations, illustrating meaningful internal reasoning.

**Weaknesses:**

1. The model uses its own predicted action distributions as inputs (queries and keys) in the attention mechanism. While this design enhances semantic reasoning, inaccurate early predictions could propagate errors through time due to the recursive dependency. The manuscript introduces an exponential moving average to smooth the predicted distributions and stabilize temporal dynamics. However, EMA may only partially mitigate the potential error accumulation caused by this recursive dependence rather than resolving it. It would be helpful for the authors to clarify whether any mechanism is explicitly designed to address this issue, or if they have other considerations regarding how such recursive errors are handled during training.
2. The authors explicitly mention that AGA treats past actions as an “unordered set.” Without explicit temporal or positional encoding, the model might struggle to capture fine-grained causal order, which could limit performance in tasks where action sequences are strictly ordered.
3. On datasets such as EPIC-Kitchens-100, the class distribution is highly imbalanced. Since the model uses predicted action distributions as semantic inputs, it would be helpful to clarify whether this design could potentially amplify frequent-class bias or, conversely, help mitigate it through semantic attention. But the reported tail-class gains remain unexplained, and there is no discussion about this aspect. Please discuss why AGA helps rare classes.
4. The paper does not report quantitative efficiency statistics. Providing these measurements or comparisons with existing approaches would strengthen the empirical analysis and clarify the computational cost of AGA.
5. The evaluation focuses exclusively on egocentric kitchen datasets (EPIC-Kitchens and EGTEA), which limits the demonstrated generality of the proposed approach. It would be valuable to verify whether the Action-Guided Attention mechanism generalizes to more open or diverse domains.
6. The ablation study could be more fine-grained. While the paper evaluates the effect of the EMA coefficient and the presence of the gating module, it lacks comparisons between different gating architectures.

**Questions:**

1. The paper states that past actions are treated as an unordered set. How does the model still maintain temporal consistency?
2. Were any techniques applied to reduce frequent-class bias? How might uncalibrated probabilities affect the quality of semantic attention on long-tail classes? If none were used, could you discuss why AGA still improves tail classes and whether calibration might further help?
3. AGA is tested with both TSN and Swin-B backbones. Did the authors observe any difference in how semantic attention behaves with different backbone architectures? For instance, does a stronger backbone reduce the relative performance gain of AGA, or does the improvement remain consistent?

---

> ### Author Response · Authors · 2025-11-25
> **Response to Reviewer XXT2 (Part 1)**
>
> We respond to the questions, weaknesses and comments by the reviewer:
>
> $${}$$
>
> **Question 1**
> > The paper states that past actions are treated as an unordered set. How does the model still maintain temporal consistency?
>
> Temporal consistency gets injected into the prediction, and consequently into the queue, through the EMA in the query.
>
> $${}$$
>
> **Question 2**
> > Were any techniques applied to reduce frequent-class bias?
>
> The loss function includes reweighting, which increases the weight for less frequent classes (as in AVT). We will incorporate this missing detail into the main paper in the experimental setup section.
>
> $${}$$
>
> **Question 3**
> > AGA is tested with both TSN and Swin-B backbones. Did the authors observe any difference in how semantic attention behaves with different backbone architectures?
> For instance, does a stronger backbone reduce the relative performance gain of AGA, or does the improvement remain consistent?
>
> We conducted an experiment on the EK100 validation set which contrasts using AGA to using frame embeddings as query and keys for both Swin-B and TSN backbone. The result shows that the Swin-B backbone experiences a degradation from 18.8 to 17.7. On the other hand, TSN drops from 17.5 to 17.0. This suggests that the relative improvement from AGA is influenced by the backbone characteristics. We hypothesize that this difference stems from the architectural properties of the two backbones: TSN is convolution-based, whereas Swin-B is attention-based. AGA benefits both, but the magnitude of the gain differs because of the semantic granularity. Despite these differences, AGA consistently improves performance across all tested backbones.
>
> $${}$$
>
> **Weakness 1**
> > The model uses its own predicted action distributions as inputs (queries and keys) in the attention mechanism. While this design enhances semantic reasoning, inaccurate early predictions could propagate errors through time due to the recursive dependency.
>
> We conducted two experiments to find out how the model copes with errors (Experiments 1 and 5 in the General Response) .
>
> With the first (Experiment 1) we wanted to explore AGA’s robustness and the impact of self-prediction accuracy. We essentially replaced self-prediction with ground truth.
>
> With the second (Experiment 5) we wanted to assess how this information loss degrades its accuracy. To this end we randomly reset the model’s self-predictions to a uniform distribution.
>
> Details, including measurements, are to be found in the section General Comments.
>
> $${}$$
>
> **Weakness 2**
> > The authors explicitly mention that AGA treats past actions as an “unordered set.” Without explicit temporal or positional encoding, the model might struggle to capture fine-grained causal order, which could limit performance in tasks where action sequences are strictly ordered.
>
> For a response, please refer to our answer to Question 1 above.
>
> $${}$$
>
> **Weakness 3**
> > On datasets such as EPIC-Kitchens-100, the class distribution is highly imbalanced. Since the model uses predicted action distributions as semantic inputs, it would be helpful to clarify whether this design could potentially amplify frequent-class bias or, conversely, help mitigate it through semantic attention. But the reported
> tail-class gains remain unexplained, and there is no discussion about this aspect. Please discuss why AGA helps rare classes.
>
> The loss function includes reweighting which increases the weight for less frequent classes (as in AVT). We will incorporate this description into the manuscript in the experimental setup section.
>
> Additionally, as shown in the table in the response to Weakness 1, AGA leverages the changes of self-prediction over the different timesteps, which is helpful to capture intrinsic action dynamics, improving the accuracy of identifying the top-5 options in tail cases.
>
> $${}$$
>
> **Weakness 4**
> > The paper does not report quantitative efficiency statistics. Providing these measurements or comparisons with existing approaches would strengthen the empirical analysis and clarify the computational cost of AGA.
>
> These points are also addressed with experiments reported in the General Comments. Experiment 3 compares the computational cost, measured in total number of FLOPs, of AVT and AGA. Experiment 4 compares the inference time of the two models.
>
> $${}$$
>
>
>
> For further responses see Response to Reviewer XXT2 (Part 2)

---

> > ### Author Response · Authors · 2025-11-25
> > **Response to Reviewer XXT2 (Part 2)**
> >
> > **Weakness 5**
> > > The evaluation focuses exclusively on egocentric kitchen datasets (EPIC-Kitchens and EGTEA), which limits the demonstrated generality of the proposed approach. It would be valuable to verify whether the Action-Guided Attention mechanism generalizes to more open or diverse domains.
> >
> > We appreciate the reviewer’s comment on evaluating more diverse domains. The recent studies are mostly egocentric, and EPIC-Kitchens and EGTEA are widely adopted benchmarks for anticipation. Extending AGA to broader, open-domain datasets is a promising avenue, and we acknowledge it as valuable suggestion for future work.
> >
> > $${}$$
> >
> > **Weakness 6**
> > > The ablation study could be more fine-grained. While the paper evaluates the effect of the EMA coefficient and the presence of the gating module, it lacks comparisons between different gating architectures.
> >
> > We thank the reviewer for this insightful suggestion. Despite its simplicity, the proposed gating module consistently improves performance across datasets, demonstrating that even a minimal gating mechanism is effective for balancing the past and current information. We consider investigating more sophisticated gating variants a valuable direction for future research building on the novel architectural design contribution proposed in this paper.

---

### Official Review · Reviewer_f8ru · 2025-10-31

**Soundness:** 3
**Presentation:** 2
**Contribution:** 3
**Rating:** 6
**Confidence:** 4

**Summary:**

This paper proposes Action-Guided Attention (AGA), a novel attention mechanism for video action anticipation that uses predicted action probabilities as queries and keys, rather than pixel-level features. The core insight is that action-level semantics provide more effective guidance for attention when dealing with the inherent uncertainty and visual clutter in anticipation tasks. The method includes an adaptive gating mechanism that balances historical context and current evidence, and enables post-training analysis via forward and backward analysis techniques. The paper demonstrates strong competitive performance on EPIC-Kitchens-100 and other benchmarks, with clear ablation studies and strong generalisation.

**Strengths:**

Overall, this paper presents a novel, well-motivated approach to action anticipation, achieving competitive results and valuable interpretability.

1. The core idea of using action-level semantics to guide attention is conceptually sound and addresses a real limitation of existing methods. The experimental validation is solid, with strong generalisation demonstrated across multiple benchmarks.

2. The interpretability analysis offers valuable insights into model behaviour, with forward analysis revealing attention patterns and backward analysis providing counterfactual reasoning paths.

3. The paper demonstrates competitive performance on EPIC-Kitchens-100 (17.5% MT5R with TSN backbone, 18.8% with Swin-B on validation), with test set performance of 16.9% MT5R (Swin-B), showing strong generalisation with a narrow validation-to-test gap (approximately 1.9 percentage points).

4. The ablation studies provide clear evidence for the contribution of both action-guided attention and adaptive gating components: replacing standard causal attention with action-guided attention provides substantial gains (15.9% to 18.2% MT5R), and adaptive gating adds further improvements (18.2% to 18.8%).

5. The method is also validated on additional benchmarks (EPIC-Kitchens-55 and EGTEA Gaze+), demonstrating robustness across different datasets and annotation regimes, including sparse supervision settings where the model must rely entirely on its own predictions.

**Weaknesses:**

1.  Theoretical grounding: The paper provides strong intuitive motivation for using action-level semantics over pixel-level features, but lacks a deeper theoretical analysis. A more formal justification would significantly strengthen the contribution.
2.  Backward analysis convergence: In Section 3.4, the convergence criterion for the gradient descent in the backward analysis is stated imprecisely as "until the sequence doesn't change any more."
3.  Ema parameter insight: While Table 6 provides a thorough empirical evaluation of the EMA parameter α, the paper lacks insight into why α=0.8 is optimal.
4.  Statistical significance and reproducibility: The paper doesn't report standard deviations, confidence intervals, or the number of runs. Providing statistical significance testing for the key comparisons (e.g., the improvements from ablations and the validation-test gap) is crucial.
5.  Missing experimental details: The size of the FIFO queue 'S' is mentioned as a hyperparameter but not specified in the experimental setup.
6. Furthermore, since AGA relies on self-predicted actions, a more explicit analysis of how prediction errors propagate and how the model handles error accumulation over time would be valuable.
7.  Failure cases and limitations: The paper doesn't discuss limitations or failure cases.

Minor comments:

*   Table organisation: Reorganising results tables to explicitly group RGB-only comparisons versus multi-modal/ensemble methods would improve clarity.
*   Related work: The related work section could better position AGA relative to methods that also use semantic or high-level features (e.g., S-GEAR). How does AGA differ from these approaches?
*   Notation and presentation: There are minor inconsistencies in notation (e.g., line 147) and formula placeholders that should be corrected for the final version.

**Questions:**

1. What is the exact stopping criterion (e.g., L2 norm threshold, maximum iterations)? How sensitive are the results to the step size η and this stopping criterion?

2. Concerning the EMA parameter: what does α=0.8  as an optimal parameter mean in terms of the effective temporal window or the balance between recent and historical information? Is this "memory buffering" optimal value dataset-dependent?

3.  Computational cost and error propagation: How do the computational cost and inference time of AGA compare to baselines?

4. Initialisation and relationship to avt: Two points of clarification: (a) At the start of a sequence, how are queries and keys initialised before the queue is filled? (b) A more detailed architectural comparison clarifying the exact relationship between AGA and the AVT baseline would be helpful.

5. When does AGA perform poorly? Are there specific types of actions or scenarios where the approach struggles?

6. What are the theoretical conditions under which action-guided attention should be more effective?

7. How does this choice relate to information-theoretic principles or representational learning theory?

---

> ### Author Response · Authors · 2025-11-25
> **Response to Reviewer f8ru (Part 1)**
>
> We respond to the questions, weaknesses and comments by the reviewer:
>
> $${}$$
>
> **Question 1**
> > What is the exact stopping criterion for backward analysis? How sensitive are the results to the step size $\eta$ and this stopping criterion?
>
> The process stops when the loss function plateaus, that is, the loss change falls below a threshold $\epsilon$, or when the maximum number of iterations steps $iter$ has been exceeded. Formally, when $| L_t(Y^j) - L_t(Y^{j-1}) | < \epsilon$ or $j \geq iter$.
>
> For the examples reported in the paper, we used a step size of $\eta = 1e2$ and a stopping criterion with $\epsilon = 1e-6$ and $iter = 5000$. When applying these settings to EK100, in 30 validation samples, the method consistently found that counterfactual actions within the top-10 predictions became the first choice. We further varied the step size from $1e-1$ to $1e5$ and found that the same result was consistently achieved when the step size lay between $1e1$ and $1e4$, using the same stopping criterion. Thus, on EK100 the results turned out to be relatively insensitive to the step size.
>
> $${}$$
>
> **Question 2**
> > Concerning the EMA parameter: what does α=0.8 as an optimal parameter mean in terms of the effective temporal window or the balance between recent and historical information? Is this "memory buffering" optimal value dataset-dependent?
>
> Intuitively, EMA injects temporal context into the query and to smooth out model prediction jitter. The optimal smoothing constant $\alpha$ depends on factors such as the video sampling rate and the target action duration. As reported in Table 6 of the paper, we found in the EK100 experiment that the MT5R was between 18.4 and 18.8 for $0.2 \leq \alpha \leq 0.8$. As this showed that the accuracy was relatively stable for $\alpha$ in that range, we applied the same value to other data sets.
>
> Regarding the temporal window, a parameter $\alpha = 0.8$ means that a weight of 0.8 is given to the last prediction, a weight of $(1-\alpha)\alpha = 0.16$ to penulatimate one, and a weight of $(1-\alpha)^2\alpha = 0.032$ to the one before the penultimate. Clearly, with $\alpha = 0.8$ the temporal context is very small. Still, such an EMA is superior to a choice of $\alpha = 1$, which leads to an MT5R of 18.2.
>
> We are currently running experiments to determine the optimal $\alpha$ for EK55 and EGTEA Gaze+, which we will report  once the experiments are finished.
>
>
> $${}$$
>
> **Question 3**
> > Computational cost and error propagation: How do the computational cost and inference time of AGA compare to baselines?
>
> To answer the question about the computational cost and inference time, we conducted two experiments where we compared AGA and AVT. They are reported as Experiments 3 and 4 in the General Response. Please, refer to that part of our response for an answer.
>
> To study how errors in the prediction affect the accuracy of AGA, we ran an experiment where we replaced a randomly increasing subset of the predictions $\hat y$ by uniform distributions, which are meaningless for the attention mechanism. The experiment and its outcome are reported as Experiment 5 in the General Response.
>
> $${}$$
>
> **Question 4**
> > Two points of clarification: (a) At the start of a sequence, how are queries and keys initialised before the queue is filled? (b) A more detailed architectural comparison clarifying the exact relationship between AGA and the AVT baseline would be helpful.
>
>
> (a) For the first frame, when the queue is empty, the attention module is bypassed and its output is initialized to a zero vector. Starting from the second frame, the queue contains at least one timestep for both Q and K, allowing the attention mechanism to begin processing information even if the queue is not yet full. We will clarify this in the paper.
>
> (b) AVT derives its queries and keys directly from frame features, which inevitably encode many low-level visual details that are not predictive of future actions. In contrast, our approach replaces frame-derived Q/K with Q/K computed from the model’s previous action prediction, which serves as a semantically aligned representation of the ongoing action dynamics. This design is intended to more effectively filter out irrelevant appearance information and to focus the attention mechanism on transitions that matter for anticipation. We observed that using prediction-based Q/K yields more stable attention and improved anticipation accuracy, supporting the intuition that action-centric signals provide a more targeted basis for temporal reasoning than frame features.
>
> $${}$$
>
> For further responses see Response to Reviewer f8ru (Parts 2 and 3)

---

> > ### Author Response · Authors · 2025-11-25
> > **Response to Reviewer f8ru (Part 2)**
> >
> > **Question 5**
> > > When does AGA perform poorly? Are there specific types of actions or scenarios where the approach struggles?
> >
> > We conducted an analysis on the EK100 validation set and collected accuracy statistics for all action, verb, and noun classes. However, no clear pattern emerged as to the scenarios in which the model falls short. We calculated Pearson's correlation between the accuracy and occurrence frequency of the three groups of action, verb, and noun classes. This results in 0.0841 for action, -0.0357 for verbs, and -0.0344 for nouns, showing that the model performance also not correlates with the action frequency.
> >
> > Section D of the Supplementary Material contains a qualitative analysis of success and failure cases on EPIC-Kitchens-55/100 and EGTEA Gaze+. From each of the three sets we selected four clips where one of the top-5 predicted actions is the ground truth, and four where none of them is the ground truth. We found that most failures occurred when key objects were occluded or when observation clues were insufficient or misleading. Such failure modes are dataset-specific and intrinsic in the data.
> >
> > Although we carefully analyzed the cases, we could not identify specific types of actions or scenarios where the model falls short and whose failure could be attributed to specific model design choices in AGA.
> >
> > $${}$$
> >
> > **Question 6**
> > > What are the theoretical conditions under which action-guided attention should be more effective?
> >
> > Providing formal theoretical guarantees for AGA is challenging. Nevertheless, we can offer an intuitive explanation of when AGA is expected to be particularly effective. AGA conditions the attention computation on the model’s own action predictions rather than on raw frame features; therefore, presumably it is less sensitive to visual clutter or scene elements unrelated to the underlying activity. In scenarios where the visual input contains distracting objects, background motion, or fine-grained variations not semantically tied to the action, we expect AGA’s prediction-guided queries and keys to focus the model on activity-relevant information, making the mechanism more robust than frame-feature based causal attention.
> >
> > $${}$$
> >
> > **Question 7**
> > > How does this choice relate to information-theoretic principles or representational learning theory?
> >
> > In general, the higher levels of a network contain more abstract information than the lower layers. In AGA, the predictions are produced by the top layer. For this reason, we use the predictions as query and keys to capture the semantics of the previous frames and extract relevant frames from the past.
> >
> > $${}$$
> >
> > **Weakness 2**
> > > Backward analysis convergence: In Section 3.4, the convergence criterion for the gradient descent in the backward analysis is stated imprecisely as "until the sequence doesn't change any more."
> >
> > We have made this precise in the answer to your Question 2 above. We will include that in the final version of the paper.
> >
> > $${}$$
> >
> > **Weakness 3**
> > >Ema parameter insight: While Table 6 provides a thorough empirical evaluation of the EMA parameter α, the paper lacks insight into why α=0.8 is optimal.
> >
> > We have tried to give a preliminary response in Question 2. Experiments for a more comprehensive answer are under way.
> >
> >
> > $${}$$
> >
> > **Weakness 4**
> > > Statistical significance and reproducibility: The paper doesn't report standard deviations, confidence intervals, or the number of runs. Providing statistical significance testing for the key comparisons (e.g., the improvements from ablations and the validation-test gap) is crucial.
> >
> >
> >
> > We followed the common standard in reporting results of experiments with Deep Learning models, where each training requires massive computing resources. Computing variances and performing significance tests would require many training runs.
> >
> > $${}$$
> >
> > **Weakness 5**
> > > Missing experimental details: The size of the FIFO queue 'S' is mentioned as a hyperparameter but not specified in the experimental setup.
> >
> > We ran experiments on EK100 where we set the queue size to $4$ and $30$, in addition to the ones with size $16$ descriebed in the paper. They are reported in the General Response as Experiment 2. Please, refer to that part for an answer.
> >
> > $${}$$
> >
> > **Weakness 6**
> > > Furthermore, since AGA relies on self-predicted actions, a more explicit analysis of how prediction errors propagate and how the model handles error accumulation over time would be valuable.
> >
> > Experiment 5 in the General Response addresses this point. Please, refer to that part of our response.
> >
> > $${}$$
> >
> > **Weakness 7**
> > > Failure cases and limitations: The paper doesn't discuss limitations or failure cases.
> >
> > We have tried to address AGA's limitations in the response to Question 5.

---

> > > ### Author Response · Authors · 2025-11-25
> > > **Response to Reviewer f8ru (Part 3)**
> > >
> > > **Comment 1**
> > > >Table organisation: Reorganising results tables to explicitly group RGB-only comparisons versus multi-modal/ensemble methods would improve clarity.
> > >
> > > Table 1 is sorted according to year of publication. Organization wrt RGB may not give much insight, since there are only two RGB-only models. It is not straightforward to draw the line between ensemble and non-ensemble approaches: The results for AVT in the leaderboard have been obtained by ensembling AVT models trained with different modalities, other models use multiple backbones or average over several runs with slightly different inputs.
> > >
> > > $${}$$
> > >
> > > **Comment 2**
> > > > Related work: The related work section could better position AGA relative to methods that also use semantic or high-level features (e.g., S-GEAR). How does AGA differ from these approaches?
> > >
> > > S-GEAR's contribution lies in aligning and establishing a semantic prototype by jointly embedding the action and language representation, while AGA  introduces an attention mechanism that conditions on the prior action space predictions to anticipate future actions.
> > >
> > > $${}$$
> > >
> > > **Comment 3**
> > > > Notation and presentation: There are minor inconsistencies in notation (e.g., line 147) and formula placeholders that should be corrected for the final version.
> > >
> > > We did not notice any inconsistencies. We would be grateful to the reviewer if he/she points them out to us.

---

### Official Review · Reviewer_3ZKi · 2025-11-05

**Soundness:** 2
**Presentation:** 3
**Contribution:** 3
**Rating:** 4
**Confidence:** 4

**Summary:**

The paper proposes Action-guided attention (AGA) for addressing the task of video action anticipation. Instead of relying purely on visual features, AGA considers a transformer-based architecture that makes use of the estimated action labels' sequence. Specifically, queries are computed based on the exponential moving average of all previous actions, while keys are constructed based on the estimated action labels within a temporal window $ S $. Values are based on the frame embeddings computed based on a frozen frame-based backbone network. The paper also proposes an adaptive gating mechanism that adaptively fuses past and current evidence.

**Strengths:**

- The idea of using a transformer-based architecture that considers the sequence of the observed action labels, in combination with the visual features, is interesting and reasonable. Surprisingly, except maybe from [R1], which follows a similar approach for long-term action anticipation, to my knowledge this direction has not received much attention yet.
- The adaptive gating mechanism is also a reasonable addition, considering that the duration of the actions varies significantly and that the proposed method does not explicitly handle the estimation of the action duration.
- The proposed method shows improved performance regarding several strong baseline methods on three widely used egocentric action anticipation datasets. It would be interesting to discuss also the suitability of the proposed method to third-person and/or 3D (Mocap) action anticipation tasks.

[R1] Gong et al., "Future transformer for long-term action anticipation", CVPR 2022.

**Weaknesses:**

- There are some important clarity issues that make the comprehension of some crucial methodological aspects challenging. Specifically, there is some confusion between the use of the terms "timesteps" and "frames", as two separate notations are used ($ i $ and $ t $). The ambiguity is also because these two indices seem to refer to different quantities, i.e. $ i $ as frame sequence index and $ t $ as timestep (e.g., in seconds similarly to $\tau_{\alpha}$).
- The text does not explicitly discuss how anticipation is performed. The ambiguity of how the anticipation is performed is closely related to the issue discussed in the previous point. For instance, in case $ t $ refers to seconds, it is not clear how the embeddings $ e_t $ are obtained from *multiple* frames that correspond to the interval between $ t-1 $ and $ t $. On the other hands, if $ t $ also refers to the frame sequence index (most probably), it is not clear how anticipation is performed as $\tau_{\alpha}$ corresponds to *multiple* steps ahead, which is not explicitly discussed in the text.
- Some additional experimental analysis would be helpful. This includes the length of the queue size $S$. It would also be interesting to show how the proposed method's performance changes when the ground-truth action labels are used for anticipation.

Minor comments
--------------
- It would be helpful to include a list of contributions in the introduction.
- L.319-321: "Nevertheless, empirical results show that AGA maintains strong performance, even under constrained annotation availability and sparse supervision.". It would be interesting to provide more details.

**Questions:**

- Can the authors provide clarifications regarding the timesteps/sequence notation and the way anticipation is achieved?
- Have the authors considered the use of AGA for third-person and/or 3D (Mocap) action anticipation tasks?
- How is the performance influenced if ground-truth action labels are used?

---

> ### Author Response · Authors · 2025-11-25
> **Response to Reviewer 3ZKi (Part 1)**
>
> We respond to the questions, weaknesses and comments by the reviewer:
>
> $${}$$
>
> **Question 1**
> > Can the authors provide clarifications regarding the timesteps/sequence notation and the way anticipation is achieved?
>
> In fact, the relationship between time and sequences was not clear. To fix this, we now distinguish explicitly between indexes and corresponding time points. The indexes arise from sampling with an interval $\Delta t$ (corresponding to a sampling rate $1/{\Delta t}$). The frame with index $t$ therefore corresponds to the time $t\cdot\Delta t$. We are also distinguishing the prediction interval $\tau_a$ and the offset $t_a$, where $t_a$ is chosen such that $\tau_a = t_a\cdot\Delta t$ holds. Predicting the action $\tau_a$ seconds ahead then corresponds to predicting the action in the frame $x_{t+t_a}$.
>
> Below is the new formulation that will replace the previous introduction of the notation:
>
> $${}$$
>
> Let $(x_t)_{t\geq 0}$  be a sequence of video frames sampled every $\Delta t > 0$ seconds, where $x_t \in \mathbb{R}^{C \times H \times W}$. For indices $r \le s$, we write
>
> $x_{r:s}$ to denote the subsequence $(x_{\max(0,r)}, \ldots, x_s)$, that is, values with $r < 0$ are clipped at zero.  Here, $t$ denotes the discrete frame index, corresponding to real time $t\cdot\Delta t$. Given the observed frames $x_{0:t}$,  the goal is to predict the future action  $y_{t + t_a} \in \mathbb{R}^{N_c}$, represented as a one-hot vector over $N_c$ classes, which occurs $t_a\cdot\Delta t = \tau_a$ seconds after the last observed frame. For each index $t$, the model outputs a probability distribution $\hat{y}_t \in [0,1]^{N_c}$ as an estimate of the true future action
>
> ${y_{t + t_a}}$.
>
> $${}$$
>
> Indexing and wallclock time are brought together in Section 4, on the experimental results, where the interval $\Delta t$ (or the correspoding sampling rate) have to be chosen to fit the prediction interval $\tau_a$. Since we applied protocols similar to those in previous research, we used values of $\tau_a = 1s$ for the EPIC-Kitchens data sets together with $\Delta_t = 1s$, and $\tau_a = 0.5s$ together with $\Delta_t = 0.5s$ for EGTEA Gaze+.
>
> Below is the new formulation in the introductory paragraphs of the section on experiments:
>
> $${}$$
>
> Experiments were conducted on three benchmarks. \EPIC-Kitchens-100 (EK100) contains 100 hours of video with 3,806 actions, including 67,217 training and 9,668 validation segments, and was evaluated at $\tau_a=1\text{s}$ and $\Delta t = 1\text{s}$. EPIC-Kitchens-55 (EK55) contains 55 hours of video with 2,513 actions, including 23,492 training and 4,979 validation segments, and was evaluated using Top-1 and Top-5 accuracy as well as MT5R at $\tau_a=1\text{s}$ and $\Delta t = 1\text{s}$. EGTEA Gaze+ comprises 28 hours of egocentric video with 106 actions, including 8,299 training and 2,022 validation clips, where we followed the protocol in (Girdhar 2021) and reported Top-1 accuracy and mean Top-1 recall at $\tau_a=0.5\text{s}$ and $\Delta t = 0.5\text{s}$.
>
> $${}$$
>
> **Question 2**
> > Have the authors considered the use of AGA for third-person and/or 3D (Mocap) action anticipation tasks?
>
> Since most of the recent action anticipation datasets are egocentric and most action anticipation research has addressed the egocentric perspective, we have done so too, to compare our approach with other work.
>
> $${}$$
>
> **Question 3**
> > How is the performance influenced if ground-truth action labels are used?
>
> Following your suggestions we conducted Experiment 1, reported in the General Response. Please, refer to that part of our response for an answer.
>
> $${}$$
>
> **Weaknesses 1 and 2**
>
> Your comments referred to the way in which we indexed frames and denoted timesteps. For a response, please have a look at our answer to Question 1 above. Regarding the way in which embeddings $e_t$ are obtained: each $e_t$ represents a single frame rather than multiple frames.
>
> $${}$$
>
> **Weakness 3**
> > Some additional experimental analysis would be helpful. This includes the length of the queue size S. It would also be interesting to show how the proposed method's performance changes when the ground-truth action labels are used for anticipation.
>
> We ran experiments on EK100 where we set the queue size to $4$ and $30$, in addition to the ones with size $16$ descriebed in the paper. They are reported in the General Response as Experiment 2. Please, refer to that part for an answer.
>
> For details on performance changes when using ground-truth action labels for anticipation, see Experiment 1 in the General Response.
>
>
> $${}$$
>
> For our last response see Response to Reviewer 3ZKi (Part 2)

---

> > ### Author Response · Authors · 2025-11-25
> > **Response to Reviewer 3ZKi (Part 2)**
> >
> > **Comment 2**
> > > L. 319-321: "Nevertheless, empirical results show that AGA maintains strong performance, even under constrained annotation availability and sparse supervision."
> > It would be interesting to provide more details.
> >
> > For EGTEA Gaze, per-frame annotations are not available; therefore, even during training the model must rely on its own self-predictions. When fine-grained supervision is available (that is, per-frame action labels), each timestep’s output can be regularized using the corresponding ground-truth action. In contrast, when such annotations are missing, the model instead learns to infer the intrinsic action transitions autonomously and is supervised only at the final timestep before the anticipation interval.

---

### Author Response · Authors · 2025-11-25
**General Response to Reviewers (Part 1)**

We thank all reviewers for their thorough and challenging reviews. They led to extensive discussions among the authors and to several new experiments.

As the next step, we will incorporate improved formulations and reports on the new experiments into the paper and the supplementary material and upload the new versions in time for feedback.

Upon acceptance we will release the complete code to ensure full transparency and reproducibility.

Some of the experiments are related to questions brought up by several reviewers. We present them here and refer to them in our responses to the individual comments.

$${}$$

### Experiment 1:  *How do Ground-truth Action Labels Influence the Performance of AGA?*

This series of experiments has been suggested by Reviewer 3ZKi.

We compared the use of the ground-truth as opposed to the self-predicted distributions $\hat{y}$ in query and keys. We represented the ground-truth with one-hot vectors. For frames without label, occurring in background our transition segments, we used one-hot vectors representing a generic "background" class.

For additional insight, we included experiments where we simplified the self-predicted distributions to one-hot vectors carrying a 1 for the action with the highest probability in $\hat{y}$,  in this way aligning the representation to the ground-truth format. For both formats, ground-truth and one-hot self-prediction, we ran two versions of the experiment: one where we used the revised format for both training and inference, and another one where we used it only for the inference.

We ran the experiments on the EK100 validation set. The results are shown in the table below. The figures refer to the accuracy on on EK100 and the columns are identical to those in Tables 1 and 2 of the paper.



| Method                                  | Train on               | Inference on           | Overall Action | Overall Noun | Overall Verb | Unseen Action | Unseen Verb | Unseen Noun | Tail Action | Tail Verb | Tail Noun |
|-----------------------------------------|------------------------|------------------------|----------------|--------------|--------------|---------------|-------------|-------------|-------------|-----------|-----------|
| GT                                      | GT                     | GT                     | 16.9           | 30.6         | 36.7         | 14.6          | 34.4        | 26.5        | 16.1        | 25.3      | 32.8      |
| GT (inference only)                     | Self-Pred              | GT                     | 17.1           | 31.9         | 36.0         | 14.1          | 32.9        | 26.3        | 16.6        | 26.8      | 32.2      |
| Top1 One-Hot Self-Pred                  | Top1 One-Hot Self-Pred | Top1 One-Hot Self-Pred | 17.2           | 31.8         | 37.7         | 14.5          | 35.6        | 27.7        | 16.3        | 26.6      | 34.2      |
| Top1 One-Hot Self-Pred (inference only) | Self-Pred              | Top1 One-Hot Self-Pred | 17.5           | 33.4         | 36.4         | 16.3          | 32.6        | 27.7        | 16.5        | 28.4      | 32.5      |
| AGA                                     | Self-Pred              | Self-Pred              | 18.8           | 32.5         | 38.7         | 16.3          | 34.4        | 28.5        | 18.4        | 27.4      | 35.0      |

The  results show that AGA, which leverages the full self-prediction distribution, achieves the highest accuracy. This can be intuitively explained by three factors:
1. The full self-prediction carries richer intrinsic information than a one-hot GT label (as further evidenced by the accuracy drop when binarizing the self-prediction into one-hot);
1. Self-prediction still provides meaningful signals when no action is annotated during the observation window (e.g., background or transition segments);
1. Although GT provides the correct top-1 future label, it lacks information about alternative plausible futures, whereas the (imperfect but semantically informative) self-prediction distribution better reflects the uncertainty and structure of the observations.

$${}$$

### Experiment 2:  *How does the Queue Size Influence the Performance of AGA?*

In the experiments on EK100 reported in the paper we set the queue length $S$ to $16$. We also conducted experiments for queue lengths $4$ and $30$. The mean top-5 recall for the different queue lengths is shown below.

| Queue Size | AGA MT5R |
|------------|----------|
| S=4        | 18.4     |
| **S=16**   | **18.8** |
| S=30       | 18.0     |

$${}$$

For the other experiments, see the General Response to Reviewers (Part 2)

---

> ### Author Response · Authors · 2025-11-25
> **General Response to Reviewers (Part 2)**
>
> ### Experiment 3:  *Computational Cost Comparison Between AVT and AGA*
>
> We computed the total number of floating-point operations (FLOPs) for both our proposed AGA and the baseline AVT.
>
> | Sequence Length | AVT (GFLOPs) | AGA (GFLOPs) |
> |-----------------|--------------|--------------|
> | 8               | 137.28       | 123.91       |
> | 16              | 274.56       | 248.29       |
> | 32              | 549.12       | 497.57       |
>
> Despite using the heavier Swin-B backbone (versus AVT’s ViT-B), AGA requires fewer FLOPs.
>
> $${}$$
>
> ### Experiment 4:  *Runtime Comparison Between AVT and AGA*
>
> We measured the inference times for AVT and AGA (with backbone Swin-B) measured on a single A100 GPU.
>
> | Sequence Length | AVT (ms) | AGA (ms) |
> |-----------------|----------|----------|
> | 8               | 14.97    | 11.4737  |
> | 16              | 20.57    | 23.4944  |
> | 32              | 33.93    | 47.3397  |
>
> The differences can be explained by the way in which the two models process the input. While AGA processes the sequence progressively, AVT handles the entire sequence in parallel.
>
> $${}$$
>
> ### Experiment 5:  *Propagation of Errors*
>
> To understand the propagation of errors, we conducted an experiment to explore how AGA degrades when subjected to inaccurate or noisy predictions that disrupt its action-guidance signal. As input, AGA received 30 frames from a 30-second clip from the EK100 validation set, with 1 second intervals in between. It had to predict the action in the last frame based on the preceding 29. For each sequence, we randomly created a permutation $\pi$ of the 30 numbers $i = 0,\ldots,29$ and performed 30 runs. In run $i$, the predictions $\hat{y}_{\pi(j)}$, where $0\leq j < i$, were forcibly reset to a uniform distribution, that is, each action was assigned the probability $\frac{1}{N_c}$. In this manner an increasing random subset of the predictions was made meaningless. In the table below, the first column contains the number of frames for which the the classifier output (i.e., $\hat{y}$) has been reset and the second column shows the resulting mean top-5 recall (MT5R).
>
> The experiment was run over the entire validation set of EK100.
>
> | #frames with randomized uniform predictions (out of 30) | AGA MT5R |
> |--------------------------|----------|
> | 0                        | 18.81    |
> | 1                        | 18.61    |
> | 2                        | 18.37    |
> | 3                        | 18.22    |
> | 4                        | 18.24    |
> | 5                        | 18.07    |
> | 6                        | 18.01    |
> | 7                        | 18.02    |
> | 8                        | 17.78    |
> | 9                        | 17.82    |
> | 10                       | 17.77    |
> | 11                       | 17.86    |
> | 12                       | 17.49    |
> | 13                       | 17.33    |
> | 14                       | 17.47    |
> | 15                       | 17.34    |
> | 16                       | 17.43    |
> | 17                       | 17.11    |
> | 18                       | 17.01    |
> | 19                       | 17.17    |
> | 20                       | 17.05    |
> | 21                       | 16.80    |
> | 22                       | 16.85    |
> | 23                       | 16.88    |
> | 24                       | 16.72    |
> | 25                       | 16.50    |
> | 26                       | 16.68    |
> | 27                       | 16.54    |
> | 28                       | 16.32    |
> | 29                       | 16.31    |
>
>
> The table shows that the accuracy steadily decreases from 18.8% to 16.3%.

---

### Author Response · Authors · 2025-12-03
**Reactions to Reviews**

Our paper presents Action-Guided Attention (AGA), a video action anticipation model based on an attention mechanism that based on past predictions generates semantic queries and keys to combine features of video frames into a new prediction.

### Our Reactions to the Reviews

The input from the reviewers led to extensive discussions among the authors and experimental investigations into the functioning of AGA, which gave us new insights and, in our view, helped us to improve both the content and the presentation of the paper.

In addition to giving some positive feedback, the reviewers asked for
 - clarifications regarding the presentation;
 - additional data and experiments to substantiate the selection of hyperparameters;
 - clarifications of the approach and comparisons with other work.

They also suggested improvements to the presentation in the paper.

In response to their reports, we
 - answered all questions and reacted to all comments;
 - reformulated those parts of the paper that had been unclear;
 - conducted experiments, both to answer questions about AGA and to compare AGA with baselines or approaches suggest by the reviewers.

We uploaded a new version of the paper, including an extended Appendix. The changed or new parts are typeset in blue.

In the next comment we describe the changes to the paper.

---

### Author Response · Authors · 2025-12-03
**Changes to the Paper**

Below, we list the changes in the paper, and explain how they respond to the reviews. The changes are of two kinds:
 - localized changes in the core paper: new paragraphs, new display formulas, reformulated sentences, references to new results in the Appendix;
 - a new section in the Appendix, with tables and figures, on the experiments inspired by the reviews.

In our original responses, we numbered for each reviewer's report the questions, weaknesses and other comments, as they appeared in the review. We use this numbering also in our explanation.

*Lines 100-107: Page 2, Related Work:*
We added references to related work on the usage of semantic labels assuggested by reviewer TEVQ's Question 3.

*Lines 140-147: Page 3, Method:*
We rewrote the paragraph in which we introduced our notation, clarifying the relation between sequence indexes and points in time, as pointed out by reviewer 3ZKi's Question 1.

*Lines 180-181: Page 4, Method:*
This is a reference to material in a later section of the paper and a later Appendix on experiments to determine optimal values for the smoothing factor $\alpha$ in the exponential moving average used to compute the query $Q_t$. This addresses Question 2 of Reviewer f8ru.

*Lines 182-185: Page 4, Method:*
Here, we describe how the queue of frame embeddings and keys is filled from the start, in response to Question 4 of Reviewer f8ru.

*Lines 216-220: Page 5, Method:*
We describe formally the last step in the computation of the prediction $\hat y$, as suggested in Question 2 of Reviewer TEVQ.

*Lines 263-268: Page 5, Method;*
*Lines 505-510: Page 10, Results:*
In these two places we make an earlier formulation concerning the convergence of the sequence $Y^{(j)}$ more precise, both regarding the formalization and the empirical observations.

*Lines 296-297: Page 6, Results;*
*Lines 690-692: Page 14, Appendix "Additional Experiments":*
We provide information on the length of the queue and the experiments by which we determined the optimal queue size, thus addressing Weakness 3 in the report of Reviewer 3ZKi.

*Lines 300-302: Page 6, Results:*
We explain how the training protocol takes account of less frequent action classes by reweighting the loss function in such cases, addressing Question 2 of Reviewer XXT2.

*Lines 303-305, 308, 310, 312-315: Page 6, Results:*
We rewrote the explanation of the parameters chosen for our experiments. We now  in the format for time and sequences introduced on Page 4 in response to Question 1 of Reviewer 3ZKi.

*Line 416: Page 8, Results:*
We refer the reader to the experiments to determine optimal values for the smoothing factor $\alpha$, suggested in Question 2 of Reviewer f8ru.

*Lines 505-510: Page 10, Results:*
Here we added details on the stopping criterion for Backward Analysis, suggested by by Reviewer XXT2, Weakness 3.

*Lines 676-818: Pages 13-16, Appendix:*
To answer the comments and questions by the reviewers, we conducted a series of new experiments, which are documented in a new section with four subsections in the Appendix (Section B, Additional Experiments). Specifically, this new material addresses the following points in the reviews:

- *B.1.1 Queue Length: addresses Weakness 3 by Reviewer 3ZKi:*
   We show that our choice of the value for the queue length $S$ is optimal in that it leads to a maximal score

- *B.1.2 Smoothing Factor: addresses Question 2 of Reviewer f8ru:*
   We show that our choice of the value for $\alpha$ is optimal.

- *B.2 Computational Cost of AGA: addresses Question 3 by Reviewer f8ru and Weakness 4 by Reviewer XXT2:*
   We compare AGA with AVT as a baseline and show that AGA's cost in FLOPs is slightly lower while the runtime is higher for larger inputs, due to the sequential nature of the computation

- *B.3 Propagation of Errors: addresses Question 2 of Reviewer f8ru:*
   We ran experiments, randomly injection errors at an increasing rate and found that the accuracy of AGA decreases proportionally to the error rate.

- *B.4 Self-Prediction vs. Ground-Truth Action: addresses Question 3 of Reviewer 3ZKi and Weakness 1 by Reviewer XXT2:*
   Since AGA uses its past predictions as queries and keys to focus attention on relevant past frame embeddings, the reviewers asked how AGA's accuracy would change if the uncertain predictions were replaced with ground truth labels. The outcome was that the accuracy decreases if the ground truth is used, thus shedding light on the functioning of the AGA mechanism.

---

### Meta-Review · Area_Chair_kc3o · 2026-01-16

**Summary:**

The paper focuses on forecasting future actions given past observations in input, and prooposes to do so by attending on predicted future actions rather than attending over pixels. This is a smart idea that was appreciated by all reviewers, and seems to be delivering good results in several leading benchmarks including EpicKitchens and EGTEA.

**Reviewer Concerns:**

The main reviewer concern was to what extend the model is robust to errors on the forecast actions, on which the model conditions to anticipate future video.  Relying on own predictions for future prediction will, in my opinion, inevitably lead to drifting. This is a well known phenomenon, eg in visual object tracking

All that said, the authors provide extra results and experiments that show that the proposed AGA outperforms the ground-truth baselines. , The fact that drifting does not appear in this setting indicates that the data is perhaps not varying enough. So, I think in the context of this paper, this result addresses the reviewer concerns sufficiently.

**Reviewer Scores:**

I cannot really answer how the review would have changed their scores, however, the authors did extra experiments and addressed all the points raised in a reasonable manner.

---

### Decision · Program_Chairs · 2026-01-26

Accept (Poster)